# HOLOGRAPHIC-(V)AE: AN END-TO-END SO(3)-EQUIVARIANT (VARIATIONAL) AUTOENCODER IN FOURIER SPACE

## ABSTRACT

Group-equivariant neural networks have emerged as a data-efficient approach to solve classification and regression tasks, while respecting the relevant symmetries of the data. However, little work has been done to extend this paradigm to the unsupervised and generative domains. Here, we present *Holographic*-(V)AE (H-(V)AE), a fully end-to-end SO(3)-equivariant (variational) autoencoder in Fourier space, suitable for unsupervised learning and generation of data distributed around a specified origin. H-(V)AE is trained to reconstruct the spherical Fourier encoding of data, learning in the process a latent space with a maximally informative invariant embedding alongside an equivariant frame describing the orientation of the data. We extensively test the performance of H-(V)AE on diverse datasets and show that its latent space efficiently encodes the categorical features of spherical images and structural features of protein atomic environments. Our work can further be seen as a case study for equivariant modeling of a data distribution by reconstructing its Fourier encoding.

## 1 INTRODUCTION

In supervised learning, the success of state-of-the-art algorithms is often attributed to respecting known inductive biases of the function they are trying to approximate. One such bias is the invariance of the function to certain transformations of the input. For example, image classification is translationally invariant. To achieve such invariance, conventional techniques use data augmentation to train an algorithm on many transformed forms of the data. However, this solution is only approximate and increases training time significantly, up to prohibitive scales for high-dimensional and continuous transformations ($\sim$500 augmentations are required to learn 3D rotation-invariant patterns (Geiger & Smidt, 2022)). Alternatively, one could use invariant features of the data (e.g. pairwise distance between different features) as input to train any machine learning algorithm (Capecchi et al., 2020; Uhrin, 2021). However, the choice of these invariants is arbitrary and the resulting network could lack in expressiveness.

Recent advances have developed neural network architectures that are equivariant under actions of different symmetry groups. These networks can systematically treat and interpret various transformation in data, and learn models that are agnostic to these transformations. For example, models equivariant to euclidean transformations have recently advanced the state-of-the-art on tasks over 3D point cloud data (Liao & Smidt, 2022; Musaelian et al., 2022; Brandstetter et al., 2022). These models are more flexible and expressive compared to their purely invariant counterparts (Geiger & Smidt, 2022), and exhibit high data efficiency.

Extending such group invariant and equivariant paradigms to unsupervised learning could map out compact representations of data that are agnostic to a specified symmetry transformation (e.g. the global orientation of an object). In recent work Winter et al. (2022) proposed a general mathematical framework for autoencoders that can be applied to data with arbitrary symmetry structures by learning an invariant latent space and an equivariant factor, related to the elements of the underlying symmetry group.

Here, we focus on unsupervised learning that is equivariant to rotations around a specified origin in 3D, denoted by the group SO(3). We encode the data in spherical Fourier space and construct holo-

grams of the data that are conveniently structured for equivariant operations. These data holograms are inputs to our end-to-end SO(3)-equivariant (variational) autoencoder in spherical Fourier space, with a fully equivariant encoder-decoder architecture trained to reconstruct the Fourier coefficients of the input; we term this approach *Holographic*-(V)AE (H-(V)AE). Similar to Winter et al. (2022), our network learns an SO(3)-equivariant latent space composed of a maximally informative set of invariants and an equivariant frame describing the orientation of the data.

We extensively test the perfomance of H-(V)AE and demonstrate high accuracy in unsupervised classification and clustering tasks for spherical images and atomic point clouds within protein structures. The learned SO(3) invariant and equivariant representations would be useful for many real world applications in computer vision and structural biology.

## 2 BACKGROUND

### 2.1 SPHERICAL HARMONICS AND IRREPS OF SO(3)

We are interested in modeling 3D data (i.e., functions in $\mathbb{R}^3$), for which the global orientation of the data should not impact the inferred model (Einstein, 1916). We consider functions distributed around a specified origin, which we express by the resulting spherical coordinates $(r, \theta, \phi)$ around the origin. In this case, the set of rotations about the origin define the 3D rotation group SO(3), and we will consider models that are rotationally equivariant under SO(3).

It is convenient to project data to spherical Fourier space to define equivariant transformations for rotations.

To map a radially distributed function $\rho(r, \theta, \phi)$ to a spherical Fourier space, we use the Zernike Fourier Transform (ZFT),

$$\hat{Z}_{\ell m}^n = \int \rho(r, \theta, \phi)\, Y_{\ell m}(\theta, \phi) R_\ell^n(r)\, \mathrm{d}V \tag{1}$$

where $Y_{\ell m}(\theta, \phi)$ is the spherical harmonics of degree $\ell$ and order $m$, where $\ell$ is a non-negative integer ($\ell \geq 0$) and $m$ is an integer within the interval $-\ell \leq m \leq \ell$. $R_\ell^n(r)$ is the radial Zernicke polynomial in 3D (Eq. A.7) with radial frequency $n \geq 0$ and degree $\ell$. $R_\ell^n(r)$ is non-zero only for even values of $n - \ell \geq 0$. Zernike polynomials - defined as the product $Y_{\ell m}(\theta, \phi) R_\ell^n(r)$ - form a complete orthonormal basis in 3D, and therefore can be used to expand and retrieve 3D shapes, if large enough $\ell$ and $n$ values are used; approximations that restrict the series to finite $n$ and $\ell$ are often sufficient for shape retrieval, and hence, desirable algorithmically. Thus, in practice, we cap the resolution of the ZFT to a maximum degree $L$ and a maximum radial frequency $N$.

The operators that describe how spherical harmonics transform under rotations are called the Wigner D-matrices. Notably, Wigner-D matrices are the irreducible representations (irreps) of SO(3), which implies that every element of the SO(3) group acting on any vector space can be represented as a direct sum of Wigner-D matrices.

As spherical harmonics form a basis for the irreps of SO(3), the SO(3) group acts on spherical Fourier space via a direct sum of irreps. Specifically, the ZFT encodes a data point into a *tensor* composed of a direct sum of *features*, each associated with a degree $\ell$ indicating the irrep that it transforms with under the action of SO(3). We refer to these tensors as SO(3)-*steerable tensors* and to the vector spaces they occupy as SO(3)-*steerable vector spaces*, or simply *steerable* for short since we only deal with the SO(3) group in this work. We note that a tensor may contain multiple features of the same degree $\ell$, which we generically refer to as distinct *channels* $c$. Throughout the paper, we refer to generic steerable tensors as $\boldsymbol{h}$ and index them by $\ell$, $m$ and $c$. We adopt the "hat" notation for individual entries to remind ourselves of the analogy with Fourier coefficients. See Figure 1A for a graphical illustration of a tensor.

### 2.2 MAPPING BETWEEN SO(3)-STEERABLE VECTOR SPACES

Constructing equivariant operations equates to constructing maps between steerable vector spaces. There are precise rules constraining the kinds of operations that guarantee a valid SO(3)-steerable output, the most important one being the Clebsch-Gordan (CG) tensor product $\otimes_{cg}$. The CG tensor

product combines two features of degrees $\ell_1$ and $\ell_2$ to produce another feature of degree $|\ell_2 - \ell_1| \leq \ell_3 \leq |\ell_1 + \ell_2|$. Let $\boldsymbol{h}_\ell \in \mathbb{R}^{2\ell+1}$ be a generic degree $\ell$ feature, with individual components $\hat{h}_{\ell m}$ for $-\ell \leq m \leq \ell$. Then, the CG tensor product is given by:

$$\hat{h}_{\ell_3 m_3} = (\boldsymbol{h}_{\ell_1} \otimes_{cg} \boldsymbol{h}_{\ell_2})_{\ell_3 m_3} = \sum_{m_1=-\ell_1}^{\ell_1} \sum_{m_2=-\ell_2}^{\ell_2} C_{(\ell_1 m_1)(\ell_2 m_2)}^{(\ell_3 m_3)} \hat{h}_{\ell_1 m_1} \hat{h}_{\ell_2 m_2} \tag{2}$$

where $C_{(\ell_1 m_1)(\ell_2 m_2)}^{(\ell_3 m_3)}$ are the Clebsch-Gordan coefficients (Tung, 1985).

## 3 HOLOGRAPHIC-(V)AE

### 3.1 SO(3) EQUIVARIANT LAYERS

**Linearity (Lin).** We construct linear layers acting on steerable tensors by learning degree-specific linear operations. Specifically, we learn weight matrices specific to each degree $\ell$, and use them to map across degree-$\ell$ feature spaces by learning linear combinations of degree-$\ell$ features in the input tensor (Section A.2.1).

**Efficient Tensor Product nonlinearity (ETP).** We utilize the CG tensor product to inject nonlinearity in the network in an equivariant way, as was originally prescribed by Kondor et al. (2018) for SO(3)-equivariant convolutional neural networks (CNNs). This type of nonlinearity enables information flow between features of different degrees, which is necessary for constructing expressive models, and injects square nonlinearity. To significantly reduce the computational and memory costs of the tensor products, we leverage some of the modifications proposed by Cobb et al. (2021). Specifically, we compute tensor products channel-wise, i.e., only between features belonging to the same channel, and we limit the connections between features of different degrees to what Cobb et al. (2021) calls the MST subset. Notably, the channel-wise computation constrains the input tensors to have the same number of channels for each feature. We found these modifications to be necessary to efficiently work with data encoded in large number of channels $C$ and with large maximum degree $L$. See Section A.2.2 for details, and Table A.7 for an ablation study showing the improvement in parameter efficiency provided by the ETP.

**Batch Norm (BN).** We normalize intermediate tensor representations degree-wise and channel-wise by the batch-averaged norms of the features, as initially proposed by Kondor et al. (2018), and do it before the ETP; see Figure 1B and Section A.2.3 for details. We found the use of this layer to speed up model convergence (Fig A.2).

**Signal Norm (SN).** It is necessary to normalize activations computed by the CG tensor product to avoid their explosion. We found using batch norm alone often caused activations to explode in the decoder during evaluation. Thus, we introduce Signal Norm, whereby we divide each steerable tensor by its *total* norm, defined as the sum of the norms of each of the tensor's features, and apply a degree-specific affine transformation ($w_\ell$) for added flexibility. Formally, the total norm for an individual tensor $h$ is computed as:

$$N_{tot} = \sum_\ell \frac{\sum_c \sum_{m=-\ell}^{\ell} (\hat{h}_{\ell m}^c)^2}{2\ell + 1} \tag{3}$$

and the features are updated as $\overline{\hat{h}_{\ell m}^c} = \hat{h}_{\ell m}^c w_\ell / \sqrt{N_{tot}}$. We note that each pre-affine normalized tensor has a total norm of 1, thus constraining the values of the individual features. Signal Norm can be seen as a form of Layer Norm that respects SO(3) equivariance (Ba et al., 2016).

**Clebsch-Gordan block (CG bl.)** We construct equivariant blocks, which we term Clebsch-Gordan blocks, by stacking together the equivariant layers introduced above, as shown in Figure 1B. Each block can take a steerable tensor of arbitrary size composed of multiple features of arbitrary degrees, and can output a steerable tensor of arbitrary size with multiple features of arbitrary degrees, by following the sparsity of the CG tensor product ($|\ell_2 - \ell_1| \leq \ell_3 \leq |\ell_1 + \ell_2|$). Crucially, if the maximum feature degree in the input tensor is $\ell_{\max}$, then the maximum feature degree that can be generated is $2\ell_{\max}$, achieved by combining two features of degree $\ell_{\max}$. We employ additive skip connections, zero-padded when appropriate, to favor better gradient flow.

## 3.2 DATA NORMALIZATION

As per standard machine learning practice (Shanker et al., 1996), we normalize the data. We do this by dividing each tensor by the average square-root total norm of the training tensors, analogously to the Signal Norm. This strategy puts the target values on a similar scale as the normalized activations learned by the network, which we speculate to favor gradient flow.

## 3.3 MODEL ARCHITECTURE

H-(V)AE has a fully rotationally equivariant architecture, and learns a *disentangled* latent space consisting of a maximally informative invariant ($\ell = 0$) component **z** of arbitrary size, as well as three orthonormal vectors ($\ell = 1$), which represent the global 3D orientation of the object and reflect the *coordinate frame* of the input tensor. Crucially, the disentangled nature of the latent space is respected at all stages of training, and is guaranteed by the model's rotational equivariance. The architecture is shown in Figure 1C. The encoder takes as input a steerable tensor with maximum degree $\ell_{\max} = L$ and, via a stack of Clebsch-Gordan blocks, iteratively and equivariantly transfers information from higher degrees to lower ones, down to the final encoder layer with $\ell_{\max} = 1$. The frame is constructed by learning two vectors and using Gram-Schmidt to find the corresponding orthonormal basis (Schmidt, 1907). The third orthonomal basis vector is then calculated as the cross product of the first two. The decoder learns to reconstruct the input from **z** and the frame, iteratively increasing the maximum degree $\ell_{\max}$ of the intermediate representations by leveraging the CG Tensor Product within the Clebsch-Gordan blocks.

An interesting question to ask is: what does the trained decoder's output look like if the frame is held constant (e.g. equal to the identity matrix)? We experimentally find that the reconstructed elements tend to be aligned with each other and hypothesize that the model is implicitly learning to maximize the overlap between training elements, providing empirical evidence in the Appendix (Fig. A.3). We call this frame the **canonical** frame following the analogy with the canonical elements in Winter et al. (2022). We note that it is possible to rotate original elements to the canonical frame thanks to the equivalence between the frame we learn and the rotation matrix within our implementation.

Within the decoder, the maximum degree $\ell_{\max,b}$ that can be outputted by each block $b$ is constrained by the sparsity of the CG tensor product. Specifically, $\ell_{\max,b} \leq 2^b$ where $b$ ranges from 1 (first block) to $B$ (last block). Since we need to reconstruct features up to degree $L$ in the decoder, we arrive at a lower bound for the number of blocks in the decoder set by $\ell_{\max,B} \geq L$, or $B \geq \log_2 L$. In our experiments, we set $\ell_{\max,b} = \min\{2^b, L\}$ and do not let $\ell_{\max,b}$ exceed the input's maximum degree $L$. Relaxing this condition might increase the expressive power of the network but at a significant increase in runtime and memory. We leave the analysis of this trade-off to future work. For the encoder and the decoder to have similar expressive power, we construct them to be symmetric with respect to the latent space (Fig. 1C). Optionally, we apply a linearity at the beginning of the encoder and at the end of the decoder; this is required for input data that does not have the same number of channels per degree since the ETP operates channel-wise. We empirically verify the equivariance of our model up to numerical errors in Table A.8.

For H-VAE, we parameterize the *invariant* part of the latent space by an isotropic Gaussian, i.e., we learn two sets of size $z$ invariants, corresponding to means and standard deviations.

## 3.4 TRAINING OBJECTIVE

We train H-(V)AE to minimize the reconstruction loss $\mathcal{L}_{rec}$ between the original and the reconstructed tensors, and, for H-VAE only, to minimize the KL-divergence of the posterior invariant latent space distribution $q(\boldsymbol{z}|\boldsymbol{x})$ from the selected prior $p(\boldsymbol{z})$ (Kingma & Welling, 2013):

$$\mathcal{L}(\boldsymbol{x}, \boldsymbol{x}') = \alpha \mathcal{L}_{\text{rec}}(\boldsymbol{x}, \boldsymbol{x}') + \beta D_{KL}(q(\boldsymbol{z}|\boldsymbol{x})||p(\boldsymbol{z})) \tag{4}$$

We use mean square error (MSE) for $\mathcal{L}_{\text{rec}}$, which as we show in Sec. A.2.4, respects the necessary property of SO(3) pairwise invariance, ensuring that the model remains rotationally equivariant.

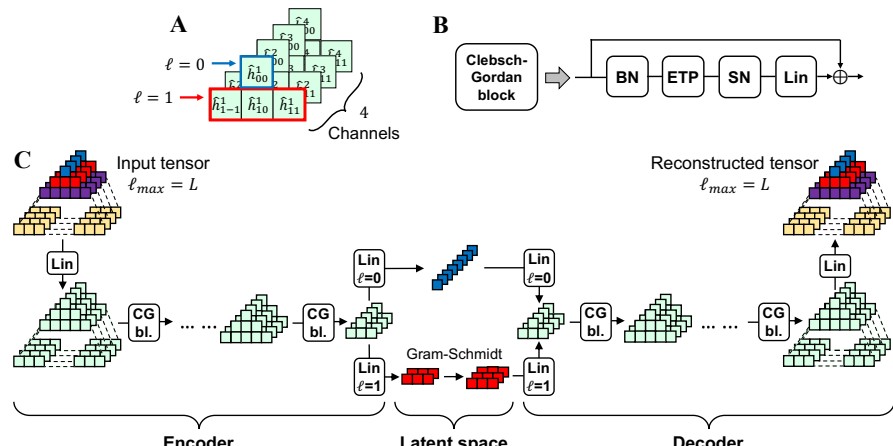

Figure 1: **Schematic of the Network architecture. A:** Schematic of a steerable tensor with $\ell_{\max} = 1$ and 4 channels per feature degree. We choose a pyramidal representation that naturally follows the expansion in size of features of higher degree. **B:** Schematic of a Clebsch-Gordan Block (CG bl.), with batch norm (BN), efficient tensor product (ETP), and signal norm (SN), and Linear (Lin) operations. **C:** Schematic of the H-AE architecture. We color-code features of different degrees in the input and in the latent space for clarity. The H-VAE schematic differs only in the latent space, where two sets of invariants are learned (means and standard deviations of an isotropic Gaussian distribution).

We cast an isotropic normal prior to the invariant latent space: $p(\boldsymbol{z}) = \mathcal{N}(\boldsymbol{0}, \boldsymbol{I})$. Hyper-parameters $\alpha$ and $\beta$ control the trade-off between reconstruction accuracy and latent space regularization Higgins et al. (2022). We find it practical to scale the reconstruction loss by a dataset-specific scalar $\alpha$ since the MSE loss varies in average magnitude across datasets. When training H-VAE, we find it beneficial to keep $\beta = 0$ for a few epochs ($E_{\text{rec}}$) so that the model can learn to perform meaningful reconstructions, and then linearly increasing it to the desired value for $E_{\text{warmup}}$ epochs to add structure to the latent space, an approach first used by Bowman et al. (2016).

## 3.5 Reconstruction assessment via cosine loss

As the scale of MSE depends on the characteristics of the data, e.g. the size of the tensors representing the data and their irreps (Fig. A.4), it is difficult to contextualize MSE values across datasets. It would be desirable to have a dimensionless metric that measures absolute "goodness" of reconstructions that is comparable across datasets. For this purpose, we propose the metric *Cosine loss* which is a normalized dot product generalized to operate on pairs of steerable tensors (akin to cosine similarity), and modified to be interpreted as a loss:

$$\text{Cosine}(\boldsymbol{x}, \boldsymbol{y}) = 1 - \frac{\boldsymbol{x} \odot \boldsymbol{y}}{\sqrt{(\boldsymbol{x} \odot \boldsymbol{x})(\boldsymbol{y} \odot \boldsymbol{y})}}, \qquad \text{with} \qquad \boldsymbol{x} \odot \boldsymbol{y} = \sum_{\ell'} (\boldsymbol{x}_{\ell'} \otimes_{cg} \boldsymbol{y}_{\ell'})_{\ell=0} \quad (5)$$

The Cosine loss is interpretable across different datasets: it is pairwise invariant, has a minimum of zero for perfect reconstructions, and has an average value of 1.0 for a pair of random tensors of any size, with a smaller variance for larger tensors (Sec. A.4). The interpretability of Cosine loss comes at the price of ignoring the relative norms of the features that are being compared, making the measure unable to reconstruct norms and thus not suitable as a training objective. However, as norms are easier to reconstruct than directions, we still find the Cosine loss useful as a noisy estimate of the model's reconstruction ability. Furthermore, Cosine loss correlates almost perfectly with MSE, especially in the mid-to-low reconstruction quality regime (SpearmanR = 0.99, Fig. A.4 and Table A.6).

## 4 RELATED WORK

**Group-equivariant neural networks.** Group-equivariant neural networks have improved the state-of-the-art on many supervised tasks, thanks to their data efficiency (Hutchinson et al., 2021; Bekkers et al., 2018; Romero & Cordonnier, 2021). Related to our work are 3D Euclidean neural networks, which are equivariant to (subsets of) the 3D Euclidean group: (Weiler et al., 2018; Brandstetter et al., 2022; Thomas et al., 2018; Batzner et al., 2022; Musaelian et al., 2022; Satorras et al., 2022; Fuchs et al., 2020; Liao & Smidt, 2022) and often use spherical harmonics and tensor products to construct SO(3) equivariant layers. Seminal work on SO(3) equivariance has been conducted for spherical images (Cohen et al., 2018; Esteves et al., 2020); we were inspired by the fully Fourier approach of Kondor et al. (2018), and leveraged operations proposed by Cobb et al. (2021).

**Equivariant representations of atomic systems.** There is a diverse body of literature on constructing representations of atomic systems that are invariant/equivariant to euclidean symmetries, leveraging Fourier transforms and CG tensor products (Drautz, 2019; Musil et al., 2021). Notably, Uhrin (2021) constructs SO(3)-invariant representations of point clouds using the ZFT and CG tensor product iterations. Our work can be seen as a data-driven instance of this framework, where we learn a compact invariant and equivariant latent space from data.

**Invariant autoencoders.** Several works attempt to learn representations that are invariant to certain classes of transformations. Shu et al. (2018) and Koneripalli et al. (2020) learn general "shape" embeddings by learning a separate "deformation" embedding. However, their networks are not explicitly equivariant to the transformations of interest. Other work proposes to learn an exactly invariant embedding alongside an approximate (but not equivariant) group action to align the input and the reconstructed data. For example, Mehr et al. (2018) learns in quotient space by sampling the group's orbit and computing the reconstruction loss by taking the infimum over the group. This approach is best suited for discrete and finite groups, and it is computationally expensive as it is akin to data augmentation. Lohit & Trivedi (2020) construct an SO(3)-invariant autoencoder for spherical signals by learning an invariant latent space and minimizing a loss which first finds the rotation that best aligns the true and reconstructed signals, introducing an added optimization step - potentially very expensive for 3D data - and reconstructing the signals only up to a global rotation. To our knowledge, the method proposed by Lohit & Trivedi (2020) is the only approach to date for unsupervised learning of non-discretized SO(3)-invariant representations. However, the rotational invariance is manually imposed by Lohit & Trivedi (2020), which is distinct from our approach that is fully equivariant and only requires simple MSE for reconstruction of data in its original orientation.

**Group-equivariant autoencoders.** A small body of work focuses on developing equivariant autoencoders. Several methods construct data and group-specific architectures to auto-encode data equivariantly, learning an equivariant representation in the process (Hinton et al., 2011; Kosiorek et al., 2019). Others use supervision to extract class-invariant and class-equivariant representations (Feige, 2022). A recent theoretical work proposes to train an encoder that encodes elements into an invariant embedding and an equivariant group action, then using a standard decoder that uses the invariants to reconstruct the elements in a canonical form, and finally applying the learned group action to recover the data's original form (Winter et al., 2022). Our method in SO(3) is closely related to this work, with the crucial difference that we use an equivariant decoder and that we learn to reconstruct the Fourier encoding of data. Furthermore, our model is variational in the invariant latent space.

## 5 EXPERIMENTS

### 5.1 ROTATED MNIST ON THE SPHERE

We extensively test the performance of H-(V)AE on the MNIST-on-the-sphere dataset (Deng, 2012). Following Cohen et al. (2018) we generate a discrete unit sphere using the Driscoll-Healey (DH) method with a bandwidth (bw) of 30, and project the MNIST dataset onto the lower hemisphere. We consider two variants, NR/R and R/R, differing in whether the training/test images have been randomly rotated (R) or not (NR). For each dataset, we map the images to steerable tensors via the Zernike Fourier Transform (ZFT) with $L = 10$, and a constant radial function $R_\ell^n = 1$, resulting in tensors with 121 coefficients.

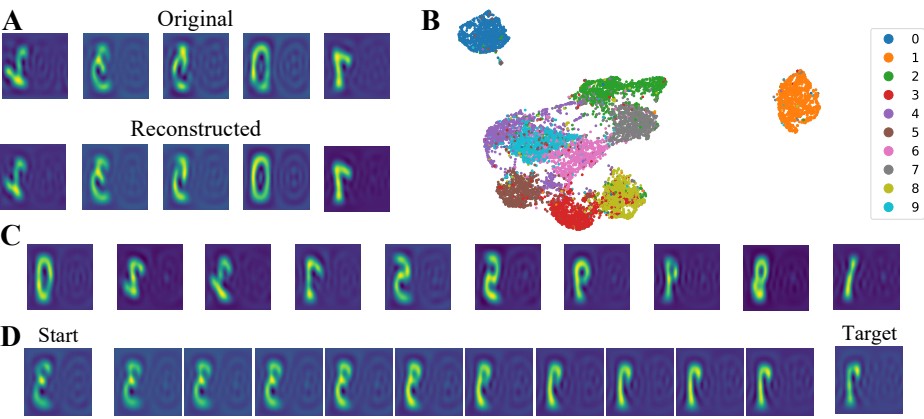

Figure 2: **H-VAE on MINST-on-the-sphere.** Evaluation on rotated digits for an H-VAE trained on non-rotated digits with $z = 16$. **A:** Original and reconstructed images in the canonical frame after inverse transform from Fourier space. The images are projected onto a plane. Distortions at the edges and flipping are side-effects of the projection. **B:** visualization of the latent space via 2D UMAP (McInnes et al., 2020). Data points are colored by digit identity. **C:** Shown are cherry-picked images generated by feeding the decoder invariant embeddings sampled from the prior distribution and the canonical frame. **D:** Example image trajectory by linearly interpolating through the learned invariant latent space. We interpolate between the learned invariant embeddings of the Start and the Target images. Then, we feed each embedding to the decoder alongside the canonical frame.

We train 8 models with combinations of the following features: training mode (NR vs. R), invariant latent space size $z$ (16 vs. 120), and model type (AE vs. VAE). In all cases the model architecture follows from Fig. 1C; see Section A.6.2 for details. All 8 models achieve very low reconstruction loss (Table 1) with no significant difference between training modes, indicating that the models successfully leverage SO(3)-equivariance to generalize to unseen orientations. Predictably, AE models have lower reconstruction loss than VAE models, and so do models with a larger latent space. Nonetheless, H-VAE achieves reliable reconstructions, as shown in Fig. 2A and Table 1. All 8 models produce an invariant latent space that naturally clusters by digit identity. We show this qualitatively for one of the models in Fig. 2B, and quantitatively by clustering the data via K-Means with 10 centroids and computing standard clustering metrics of Purity (Aldenderfer & Blashfield, 1984) and V-measure (Rosenberg & Hirschberg, 2007) in Table 1.

All 8 models achieve much better clustering metrics than Rot-Inv AE (Lohit & Trivedi, 2020), with VAE models consistently outperforming AE models. We also train a linear classifier (LC) to predict digit identity from invariant latent space descriptors, achieving comparable accuracy to Rot-Inv AE with the same latent space size. We do not observe any difference between VAE and AE models in terms of classification accuracy. Using a KNN classifier instead of LC further improves performance (Table A.2).

As H-VAE is a generative model, we generate random spherical images by sampling invariant latent embeddings from the prior distribution, and observing diversity in digit type and style (Fig. 2C and Fig. A.5). We further assess the quality of the invariant latent space by generating images via linear interpolation of the invariant embeddings associated with two test images. The interpolated images present spatially consistent transitions (Fig. 2D and Fig. A.6), which is a sign of a semantically well-structured latent space.

To understand the meaning of the learned frames, we visualize the sum of images in the canonical frame (i.e. the identity matrix). We hypothesize that H-(V)AE learns to optimally overlap the training images when in the same frame. Indeed, by visualizing the sum of the training images in the canonical frame, we can verify that images are well aligned within the same digit type and with varying degrees across different digit types depending on the content of training data (Fig. A.3).

Table 1: **Performance metrics on MNIST-on-the-sphere and Shrec17**. Reconstruction loss, clustering metrics, classification accuracy in the latent space using a linear classifier, and retrieval metrics (Shrec17 only) are shown. We only report scores presented in the corresponding papers of origin.

| Dataset | Type | Method | z | bw | Cosine | Purity | V-meas. | Class. Acc. | P@N | R@N | F1@N | mAP | NDCG |
|---|---|---|---|---|---|---|---|---|---|---|---|---|---|
| MNIST | Supervised | (Cobb et al., 2021) NR/R | - | 30 | - | - | - | 0.993 | - | - | - | - | - |
| | Unsupervised | (Lohit & Trivedi, 2020) NR/R | 120 | 30 | - | 0.40 | 0.35 | **0.894** | - | - | - | - | - |
| | | H-AE NR/R (Ours) | 120 | 30 | 0.017 | 0.62 | 0.48 | 0.877 | - | - | - | - | - |
| | | H-AE R/R (Ours) | 120 | 30 | 0.018 | 0.51 | 0.41 | 0.881 | - | - | - | - | - |
| | | H-AE NR/R (Ours) | 16 | 30 | 0.025 | 0.62 | 0.51 | 0.820 | - | - | - | - | - |
| | | H-AE R/R (Ours) | 16 | 30 | 0.024 | 0.65 | 0.52 | 0.833 | - | - | - | - | - |
| | | H-VAE NR/R (Ours) | 120 | 30 | 0.037 | 0.70 | **0.59** | 0.883 | - | - | - | - | - |
| | | H-VAE R/R (Ours) | 120 | 30 | 0.037 | 0.65 | 0.53 | 0.884 | - | - | - | - | - |
| | | H-VAE NR/R (Ours) | 16 | 30 | 0.057 | 0.67 | 0.54 | 0.812 | - | - | - | - | - |
| | | H-VAE R/R (Ours) | 16 | 30 | 0.055 | **0.72** | 0.57 | 0.830 | - | - | - | - | - |
| Shrec17 | Supervised | (Esteves et al., 2020) | - | 128 | - | - | - | - | 0.717 | 0.737 | - | 0.685 | - |
| | | (Cobb et al., 2021) | - | 128 | - | - | - | - | 0.719 | 0.710 | 0.708 | 0.679 | 0.758 |
| | Unsupervised | (Lohit & Trivedi, 2020) | 120 | 30 | - | 0.41 | 0.34 | 0.578 | 0.351 | 0.361 | 0.335 | 0.215 | 0.345 |
| | | H-AE (Ours) | 40 | 90 | 0.130 | 0.50 | 0.41 | **0.654** | **0.548** | **0.569** | **0.545** | **0.500** | **0.597** |
| | | H-VAE (Ours) | 40 | 90 | 0.151 | **0.52** | **0.42** | 0.631 | 0.512 | 0.537 | 0.512 | 0.463 | 0.568 |

## 5.2 SHREC17

The Shrec17 dataset consists of 51k colorless 3D models belonging to 55 object classes, with a 70/10/20 train/valid/test split (noa). We use the variant of the dataset where each model is perturbed by random rotations. Converting 3D shapes into spherical images preserves topological surface information, while significantly simplifying the representation. We follow Cohen et al. (2018) and project surface information from each model onto an enclosing DH spherical grid with a bandwidth of 90 via a ray-casting scheme, generating spherical images with 6 channels. We then apply the ZFT with $L = 14$ and a constant radial function $R_\ell^n = 1$ to each channel individually, resulting in a tensor with 1350 coefficients. We train an AE and a VAE model (Sec. A.6.3) and evaluate them similarly to the MNIST dataset and compute the Shrec17 retrieval metrics via the latent space linear classifier's predictions. H-AE achieves the best classification and retrieval results for autoencoder-based models, and is competitive with supervised models despite the lower grid bandwidth and the small latent space (Table 1). Using KNN classification instead of a linear classifier further improves performance (Table A.3). H-VAE achieves slightly worse classification results but better clustering metrics compared to H-AE. While reconstruction loss is low, there is still significant margin of improvement. We partially attribute Lohit & Trivedi (2020)'s low scores to the low grid bandwidth. However, we note that the size and runtime of our method does not scale with grid bandwidth, since the size of the reconstructed tensor learned by our method does not depend on it.

## 5.3 PROTEIN NEIGHBORHOODS

We test H-(V)AE on a challenging point cloud dataset, comprised of spherical atomic environments surrounding a residue within a protein structure, which we term protein *neighborhoods*. We use ProteinNet (AlQuraishi, 2019) splits to avoid any redundancy in training and test sets (Sec. A.6.5). We consider all atomic neighborhoods within a radius of 12.5Å surrounding central residue types CYS, GLU or HIS (Fig. 3A). We only consider backbone atoms ($C\alpha$, C, O and N). For each neighborhood centered at the central residue's $C\alpha$, we compute the ZFT (Eq. 1) with $L = 4$ and $N = 26$. We then concatenate features of the same degree resulting in a tensor with 1240 coefficients. The final dataset contains 303k/75k/4.4k (train/valid/test) tensors. We train an H-

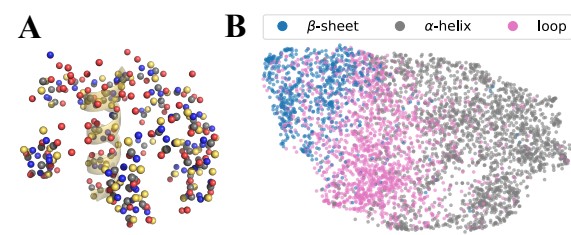

Figure 3: **H-AE on protein neighborhoods. A:** An example protein neighborhood of backbone atoms: $C\alpha$ (yellow), C (gray), N (blue) and O (red). The secondary structure of the central residue is shown overlaid in yellow. **B:** 2D UMAP visualization of the invariant latent space learned by H-AE, colored by secondary structure of the central residue.

AE model with an invariant latent space size $z$ of 64 (see Sec. A.6.5). While the test Cosine loss has still margins of improvement ($\sim 0.161$) the latent space is meaningfully organized according to relevant neighborhood statistics, such as the secondary structure of the central residue ($\sim 87\%$ KNN classification accuracy - Fig. 3B), number of atoms in the neighborhood (Fig. A.10), and neighborhood's average Solvent Accessible Surface Area (SASA), which is a proxy value for how buried the residue is in the protein (Fig. A.11). We further benchmark H-(V)AE via extensive ablation experiments on a simpler point cloud dataset consisting of individual amino acids (Sec. A.3).

## 6 USING AN UNCONSTRAINED DECODER

It is possible to merge use our SO(3)-equivariant encoder and fourier-space formulation with Winter et al. (2022)'s framework, in which the learned rotation (i.e. the frame) is applied to the output space of an unconstrained decoder instead of being fed as input to an equivariant decoder. The resulting model is theoretically equivalent to ours with regards to respecting equivariance between input and output. We implement and test a version of this model on the Rotated MNIST-on-the-sphere dataset, and show that it is slightly worse but comparable to using our fully-equivariant decoder in terms of speed (1.3x slower) and performance (0.037 vs. 0.025 Cosine loss, Table A.1). Given the similarity in performance, we favor the simplicity and elegance of our equivariant decoder, which is constructed to be symmetric to the encoder, thus automatically endowing it with similar representational power and without the need to tune an architecture made with different base components. Implementation details and extended discussion on this matter can be found in Section A.4.

## 7 PROPERTIES AND LIMITATIONS

The information content of the latent space is limited by the maximum degree $L$ and radial frequency $N$ set as cutoff for the spherical Fourier series. Such truncation is necessary for computational feasibility, but it limits resolution, evident when reconstructing a signal.

Another related limitation is the higher inaccuracy of H-(V)AE in reconstructing features associated with larger degrees (Fig. A.12), resulting in a failure to learn fine-grained details of data. This failure could explain the network's troubles on MNIST in differentiating between 4's and 9's, which tend to be hand-drawn similarly.

## 8 CONCLUSION

In this work, we develop the first end-to-end SO(3)-equivariant (V)AE, suitable for data distributed around a center. The model learns an invariant embedding describing the data in a "canonical" orientation alongside an equivariant frame describing the data's original orientation relative to the canonical one.

We use the learned invariants to achieve state-of-the-art unsupervised clustering and classification results on various spherical image datasets, and atomic environments within protein structures. By making our model variational in its invariant latent space, we enhanced the quality of clustering quality and made the model generative. Our model is defined fully in spherical Fourier space, and thus, can reach a desired expressiveness without a need for excessive computational resources.

The example of protein neighborhoods demonstrates that our model can be used to efficiently learn SO(3) invariant and equivariant representations of high-dimensional data, such as atomic environments. While in this paper we limit our analysis to backbone atoms within a small protein subspace, our method can be extended to all-atom representations of the entire protein universe.

Going forward, we expect our method will be useful in devising models for larger units from representations learned on smaller building blocks. For example, embeddings learned by H-(V)AE representing protein neighborhoods can be used to coarse-grain full atom representations of protein structures to facilitate structure-based predictions. A similar approach, albeit in the sequence domain, has been used by Omega-fold (Wu et al., 2022), where representations for amino acids in a protein from a separate language model are used as inputs to a neural network to learn an MSA-free model for protein folding. Beyond proteins, our approach could be used for other hierarchical models in large 3D structures, for which respecting rotational symmetry is desirable.

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

# A    APPENDIX

## A.1    EXPANDED BACKGROUND ON SO(3)-EQUIVARIANCE

### A.1.1    INVARIANCE AND EQUIVARIANCE

Let $f : X \rightarrow Y$ be a function between two vector spaces and $\mathfrak{G}$ a group, where $\mathfrak{G}$ acts on $X$ and via representation $\boldsymbol{D}_X$ and on $Y$ via representation $\boldsymbol{D}_Y$. Then, $f$ is said to be $\mathfrak{G}$-equivariant iff $f(\boldsymbol{D}_X(\mathfrak{g})\boldsymbol{x}) = \boldsymbol{D}_Y(\mathfrak{g})f(\boldsymbol{x})\,, \forall \boldsymbol{x} \in X \wedge \forall \mathfrak{g} \in \mathfrak{G}$. We note that invariance is a special case of equivariance where $\boldsymbol{D}_Y(\mathfrak{g}) = \boldsymbol{I}\,, \forall \mathfrak{g} \in \mathfrak{G}$.

### A.1.2    GROUP REPRESENTATIONS AND THE IRREPS OF SO(3)

Groups can concretely act on distinct vector spaces via distinct group representations. Formally, a group representation defines a set of invertible matrices $\boldsymbol{D}_X(\mathfrak{g})$ parameterized by group elements $\mathfrak{g} \in \mathfrak{G}$, which act on vector space $X$. As an example, two vector spaces that transform differently under the 3D rotation group SO(3)- and thus have different group representations - are scalars, which do not change under the action of SO(3), and 3D vectors, which rotate according to the familiar 3D rotation matrices.
A special kind of representation for any group are the irreducible representations (irreps) which are provably the "smallest" nontrivial (i.e., they have no nontrivial group-invariant subspaces) representations. The irreps of a group are special because it can be proven that any finite-dimensional unitary group representation can be decomposed into a direct sum of irreps (Tung, 1985). This applies to SO(3) as well, whose irreps are the Wigner-D matrices, which are $(2\ell + 1 \times 2\ell + 1)$-dimensional matrices, each acting on a $(2\ell + 1)$-dimensional vector space:

$$\boldsymbol{D}_\ell(\mathfrak{g}) \quad \text{for } \ell = 0, 1, 2, ... \tag{A.1}$$

Therefore, every element of the SO(3) group acting on any vector space can be represented as a direct sum of Wigner-D matrices.

### A.1.3    STEERABLE FEATURES

A G-steerable vector is a vector $\boldsymbol{x} \in X$ that under some transformation group $\mathfrak{G}$, transforms via matrix-vector multiplication $\boldsymbol{D}_X(\mathfrak{g})\boldsymbol{x}$; here, $\boldsymbol{D}_X(\mathfrak{g})$ is the group representation of $\mathfrak{g} \in \mathfrak{G}$. For example, a vector in 3D Euclidean space is SO(3)-steerable since it rotates via matrix-vector multiplication using a rotation matrix.

However, we can generalize 3D rotations to arbitrary vector spaces by employing the irreps of SO(3). We start by defining a degree-$\ell$ feature as a vector that is SO(3)-steerable by the $\ell^{th}$ Wigner-D matrix $\boldsymbol{D}_\ell$. Given the properties of irreps, we can represent any SO(3)-steerable vector as the direct sum of two or more independent degree-$\ell$ features, e.g. $\boldsymbol{x} = \boldsymbol{x}_{\ell_1} \oplus \boldsymbol{x}_{\ell_2} \oplus ... \oplus \boldsymbol{x}_{\ell_n}$. The resulting vector, which we refer to as a *tensor* to indicate that it is composed of multiple individually-steerable vectors, is SO(3)-steerable via the direct sum of Wigner-D matrices of corresponding degrees. This tensor is a block-diagonal matrix with the Wigner-D matrices along the diagonal: $\boldsymbol{D}(\mathfrak{g}) = \boldsymbol{D}_{\ell_1}(\mathfrak{g}) \oplus \boldsymbol{D}_{\ell_2}(\mathfrak{g}) \oplus ... \oplus \boldsymbol{D}_{\ell_n}(\mathfrak{g})$.

### A.1.4    SPHERICAL HARMONICS AND THE SPHERICAL FOURIER TRANSFORM

Spherical harmonics are a class of functions that form a complete and orthonormal basis for functions $f(\theta, \phi)$ defined on a unit sphere; $\theta$ and $\phi$ are the azimuthal and the polar angles in the spherical coordinate system. In their complex form, spherical harmonics are defined as,

$$Y_{\ell m}(\theta, \phi) = \sqrt{\frac{2n+1}{4\pi}\frac{(n-m)!}{(n+m)!}} e^{im\phi} P_\ell^m(\cos\theta) \tag{A.2}$$

where $\ell$ is a non-negative integer ($0 \leq \ell$) and $m$ is an integer within the interval $-\ell \leq m \leq \ell$. $P_\ell^m(\cos\theta)$ is the Legendre polynomial of degree $\ell$ and order $m$, which together with the complex exponential $e^{im\phi}$ define sinusoidal functions over the angles $\theta$ and $\phi$ in the spherical coordinate system. Spherical harmonics are used to describe angular momentum in quantum mechanics.

Notably, spherical harmonics also form a basis for the irreps of SO(3), i.e., the Wigner-D matrices. Specifically, the SO(3) group acts on the $\ell^{th}$ spherical harmonic via the $\ell^{th}$ Wigner-D matrix:

$$Y_{\ell m}(\theta, \phi) \xrightarrow{\mathfrak{g} \in SO(3)} \sum_{m'=-\ell}^{\ell} D_{\ell}^{m'm}(\mathfrak{g}) Y_{\ell m'}(\theta, \phi) \tag{A.3}$$

Therefore, any data encoded in a spherical harmonics basis is acted upon by the SO(3) group via a direct sum of the Wigner-D matrices corresponding to the basis functions being used. Using our nomenclature, any such data encoding constitutes a steerable tensor. We can thus map any function $f(\theta, \phi)$ defined on a sphere into a steerable tensor using the Spherical Fourier Transform (SFT):

$$\hat{f}_{\ell m} = \int_0^{2\pi} \int_0^{\pi} f(\theta, \phi) Y_{\ell m}(\theta, \phi) \sin \theta \, \mathrm{d}\theta \, \mathrm{d}\phi \tag{A.4}$$

The signal can be reconstructed in the real space using the corresponding inverse Fourier transform. For computational purposes, we truncate Fourier expansions at a maximum angular frequency $L$, which results in an approximate reconstruction of the signal $\tilde{f}(\theta, \varphi)$ up to the angular resolution allowed by $L$,

$$\tilde{f}(\theta, \varphi) = \sum_{\ell=0}^{L} \sum_{m=-\ell}^{\ell} \hat{f}_{\ell m} Y_{\ell m}(\theta, \phi) \tag{A.5}$$

Here, $\hat{f}_{\ell m}$ are the functions' Spherical Fourier coefficients.

### A.1.5  ZERNIKE POLYNOMIALS AND THE ZERNIKE FOURIER TRANSFORM

To encode a function $\rho(r, \theta, \phi)$ with both radial and angular components, we use Zernike Fourier transform,

$$\hat{Z}_{\ell m}^n = \int \rho(r, \theta, \phi) \, Y_{\ell m}(\theta, \phi) R_{\ell}^n(r) \, \mathrm{d}V \tag{A.6}$$

where $R_{\ell}^n(r)$ is the radial Zernike polynomial in 3D defined as,

$$R_{\ell}^n(r) = (-1)^{\frac{n-\ell}{2}} \sqrt{2n+3} \binom{\frac{n+\ell+3}{2} - 1}{\frac{n-\ell}{2}} |r|^{\ell} \, {}_2F_1\left(-\frac{n-1}{2}, \frac{n+\ell+3}{2}; \ell + \frac{3}{2}; |r|^2\right) \tag{A.7}$$

Here, ${}_2F_1(\cdot)$ is an ordinary hypergeometric function, and $n$ is a non-negative integer representing a radial frequency, controlling the radial resolution of the coefficients. $R_{\ell}^n(r)$ is non-zero only for even values of $n - \ell \geq 0$. Zernike polynomials form a complete orthonormal basis in 3D, and therefore, can be used within a Fourier transform to expand and retrieve any 3D shape, if large enough $\ell$ and $n$ coefficient are used. We refer to the Fourier transform of Eq. A.6 as the Zernike Fourier Trasform (ZFT).

To represent point clouds, a common choice for the function $\rho(\boldsymbol{r}) \equiv \rho(r, \theta, \phi)$ is the sum of Dirac-$\delta$ functions centered at each point:

$$\rho(\boldsymbol{r}) = \sum_{i \in \text{points}} \delta(\rho(\boldsymbol{r}_i) - \rho(\boldsymbol{r})) \tag{A.8}$$

This choice is powerful because the forward transform has a closed-form solution that does not require a discretization of 3D space for numerical computation. Specifically, the ZFT of a point cloud follows:

$$\hat{Z}_{\ell m}^n = \sum_{i \in \text{points}} R_n^{\ell}(r_i) Y_{\ell m}(\theta_i, \varphi_i) \tag{A.9}$$

Similar to SFT, we can reconstruct the data using inverse ZFT and define approximations by truncating the angular and radial frequencies at $L$ and $N$, respectively,

$$\tilde{\rho}(r, \theta, \varphi) = \sum_{\ell=0}^{L} \sum_{m=-\ell}^{\ell} \sum_n^N \hat{Z}_{\ell m}^n R_{\ell}^n(r) Y_{\ell m}(\theta, \varphi) \tag{A.10}$$

The use of other radial bases is possible within our framework, as long as they are complete. Orthonormality is also desirable as it ensures that each basis encodes different information, resulting

in a more efficient encoding of the coefficients. We use Zernike polynomials following Boyd & Yu (2011), which demonstrates that encoding with Zernike polynomials result in a faster convergence compared to the radial basis functions localized at different radii, as well as most other orthogonal harmonic bases, with the exception of Logan-Shepp. "Faster convergence" indicates that fewer frequencies are required to encode the same information. Uhrin (2021) also uses Zernike to construct invariant descriptors of atomic environments. Other equivariant methods use Bessel functions (Musaelian et al., 2022), though, according to Boyd & Yu (2011), Zernike encoding results in faster convergence.

## A.2 DETAILS OF H-(V)AE COMPONENTS

### A.2.1 LINEARITY

Let us consider a feature $\boldsymbol{h}_\ell$ containing $C$ features of the same degree $\ell$. $\boldsymbol{h}_\ell$ can be represented as a $C \times (2\ell + 1)$ matrix where each row constitutes an individual feature. Then, we learn weight matrix $\boldsymbol{W}_\ell \in \mathbb{R}^{C \times K}$ that linearly maps $h_\ell$ to $\overline{h}_\ell \in \mathbb{R}^{K \times (2\ell+1)}$:

$$\overline{\boldsymbol{h}}_\ell = \boldsymbol{W}_\ell^T \boldsymbol{h}_\ell \tag{A.11}$$

### A.2.2 EFFICIENT TENSOR PRODUCT (ETP)

**Channel-wise tensor product nonlinearity.** We effectively compute $C$ tensor products, each between features belonging to the same channel $c$, and concatenate all output features of the same degree. In other words, features belonging to different channels are not mixed in the nonlinearity; the across-channel mixing is instead done in the linear layer. This procedure reduces the computational time and the output size of the nonlinear layers with respect to the number of channels $C$, from $\mathcal{O}(C^2)$ for a "fully-connected" tensor product down to $\mathcal{O}(C)$. The number of learnable parameters in a linear layer are proportional to the size of the output space in the preceding nonlinear layer. Therefore, reducing the size of the nonlinear output substantially reduces the complexity of the model and the number of model parameters. This procedure also forces the input tensor to have the same number of channels for all degrees. We refer the reader to Cobb et al. (2021) for further details and for a nice visualization of this procedure.

**Minimum Spanning Tree (MST) subset for degree mixing.** To compute features of the same degree $\ell_3$ using the CG Tensor Product, pairs of features of varying degrees may be used, up to the rules of the CG Tensor Product. Specifically, pairs of features with any degree pair $(\ell_1, \ell_2)$ may be used to produce a feature of degree $\ell_3$ as long as $|\ell_1 - \ell_2| \leq \ell_3 \leq \ell_1 + \ell_2$. Features of the same degree are then concatenated to produce the final equivariant (steerable) output tensor.
Since each produced feature (often referred to as a "fragment" in the literature Kondor et al. (2018); Cobb et al. (2021)) is independently equivariant, computing only a subset of them still results in an equivariant output, albeit with lower representational power. Reducing the number of computed fragments is desirable since their computation cannot be easily parallelized. In other words, to reduce complexity we should identify a small subset of fragments that can still offer sufficient representational power. In this work we adopt the "MST subset" solution proposed by Cobb et al. (2021), which adopts the following strategy: when computing features of the same degree $\ell_3$, exclude the degree pair $(\ell_0, \ell_2)$ if the $(\ell_0, \ell_1)$ and the $(\ell_1, \ell_2)$ pairs have already been computed. The underlying assumption behind this solution is that the last two pairs already contain some information about the first pair, thus making its computation redundant.
The resulting subset of pairs can be efficiently computed via the Minimum Spanning Tree of the graph describing the possible pairs used to generate features of a single degree $\ell_3$, given the maximum desired degree $\ell_{max}$. As multiple such trees exist, we choose the one minimizing the computational complexity by weighting each edge (i.e. each pair) in the graph accordingly (edge $(\ell_1, \ell_2)$ gets weight $(2\ell_1 + 1)(2\ell_2 + 1)$). The subset is also augmented to contain all the pairs with same degree to inject more nonlinearity. This procedure reduces the complexity in number of pairs with respect to $\ell_{\max}$ from $\mathcal{O}(\ell_{\max}^2)$ - when all possible pairs are used - down to $\mathcal{O}(\ell_{\max})$. We refer the reader to Cobb et al. (2021) for more details and for a nice visualization.

### A.2.3 BATCH NORM

Let us consider a batch of steerable tensors $h$ which we index by batch $b$, degree $\ell$, order $m$ and channel $c$. During training, we compute a batch-averaged norm for each degree $\ell$ and each channel $c$ as,

$$N_\ell^c = \frac{1}{B} \sum_{b=1}^{B} \frac{1}{2\ell+1} \sum_{m=-\ell}^{\ell} (\hat{h}_{\ell m}^{cb})^2 \tag{A.12}$$

Similar to standard batch normalization, we also keep a running estimate of the training norms $N_\ell^{c,tr(i)}$ using momentum $\xi$, set to 0.1 in all our experiments:

$$N_\ell^{c,tr(i)} = \xi N_\ell^c + (1-\xi) N_l^{c,tr(i-1)} \tag{A.13}$$

We then update the features of the steerable tensor using the real batch-averaged norms during training, and the running batch-averaged norms during testing, together with a learned affine transformation:

$$\overline{\hat{h}_{\ell m}^{cb}} = \frac{\hat{h}_{\ell m}^{cb}}{\sqrt{N_\ell^c}} \, w_\ell^c \qquad\qquad \text{training} \tag{A.14}$$

$$\overline{\hat{h}_{\ell m}^{cb}} = \frac{\hat{h}_{\ell m}^{cb}}{\sqrt{N_\ell^{c,tr(i)}}} \, w_\ell^c \qquad \text{evaluation} \tag{A.15}$$

### A.2.4 PAIRWISE INVARIANT RECONSTRUCTION LOSS

To reconstruct a signal within an equivariant model it is desirable to have a *pairwise invariant* reconstruction loss, i.e., a loss $\mathcal{L}_{rec}$ such that $\mathcal{L}_{rec}(\boldsymbol{x}, \boldsymbol{y}) = \mathcal{L}_{rec}(\boldsymbol{D}(\mathfrak{g})\boldsymbol{x}, \boldsymbol{D}(\mathfrak{g})\boldsymbol{y})$ where $\boldsymbol{D}$ is the representation of the group element $\mathfrak{g}$ acting on the space that $x$ and $y$ inhabit (e.g. a rotation matrix if $\boldsymbol{x}$ and $\boldsymbol{y}$ are vectors in Euclidean 3D space, or a degree-$\ell$ wigner-D matrix if $\boldsymbol{x}$ and $\boldsymbol{y}$ are degree-$\ell$ vectors). This property is necessary for the model to remain equivariant, i.e., given that the network is agnostic to the transformation of the input under group operation $\boldsymbol{x} \to \boldsymbol{D}(\mathfrak{g})\boldsymbol{x}$ by producing a similarly transformed output $\boldsymbol{y} \to \boldsymbol{D}(\mathfrak{g})\boldsymbol{y}$, we want the reconstruction loss to be agnostic to the same kind of transformation as well

The MSE loss is pairwise invariant for any degree-$\ell$ feature on which SO(3) acts via the $\ell$'s Wigner-D matrix. Consider two degree-$\ell$ features $\boldsymbol{x}_\ell$ and $\boldsymbol{y}_\ell$ acted upon by a Wigner-D matrix $D_\ell(\mathfrak{g})$ parameterized by rotation $\mathfrak{g}$ (we drop the $\mathfrak{g}$ and $\ell$ indexing for clarity):

$$
\begin{aligned}
\mathrm{MSE}(\boldsymbol{Dx}, \boldsymbol{Dy}) &= (\boldsymbol{Dx} - \boldsymbol{Dy})^T (\boldsymbol{Dx} - \boldsymbol{Dy}) \\
&= (\boldsymbol{D}(\boldsymbol{x} - \boldsymbol{y}))^T (\boldsymbol{D}(\boldsymbol{x} - \boldsymbol{y})) \\
&= (\boldsymbol{x} - \boldsymbol{y})^T \boldsymbol{D}^T \boldsymbol{D}(\boldsymbol{x} - \boldsymbol{y}) \\
&= (\boldsymbol{x} - \boldsymbol{y})^T (\boldsymbol{x} - \boldsymbol{y}) \qquad \text{since Wigner-D matrices are unitary} \\
&= \mathrm{MSE}(\boldsymbol{x}, \boldsymbol{y})
\end{aligned}
\tag{A.16}
$$

Since the MSE loss is pairwise invariant for every pair of degree-$\ell$ features, it is thus pairwise invariant for pairs of steerable tensors composed via direct products of steerable features.

### A.2.5 COSINE LOSS

**Proof that Cosine loss is pairwise invariant.** The generalized dot product $\odot$ from Eq. 5 is pairwise invariant in the same way that the dot product between two 3D vectors depends only on their relative orientations but not the global orientation of the whole two-vector system. Therefore, the whole Cosine loss expression is pairwise invariant, since all of its components are pairwise invariant.

**On the use case of Cosine loss.** We introduce the Cosine loss as a measure of reconstruction that is both interpretable and comparable across datasets– the two characteristics that MSE does not have. A measure with these characteristics is practically useful for training of a networkd because it provides an estimate for how much better the reconstructions can get if the network's hyperparameters

were to be further optimized. For example, looking at the Cosine loss in Table A.6, we see that our model trained on Shrec17 (best Cosine = 0.130) is not as well optimized as our model trained on MNIST (best Cosine = 0.017). Using MSE, the trend is reversed ($1.8 \times 10^{-3}$ vs. $6.7 \times 10^{-3}$), since the scale of MSE depends on the size of the irreps of the data (Fig. A.4).

## A.3 TOY AMINO-ACIDS

We train H-(V)AE on single amino acids represented as atomic point clouds, extracted from structures in the Protein Data Bank (PDB) (Berman et al., 2000). We collect atomic point clouds of 50k residues from PDB evenly distributed across residue types and apply a 40/40/20 train/valid/test split. Residues of the same type have different conformations and naturally have noisy coordinates, making this problem a natural benchmark for our method.

We consider atom-type-specific clouds (C, O, N and S; we exclude H) centered at the residue's C$\alpha$ and compute the ZFT (Eq. 1) with $L = 4$ and $N = 20$ within a radius of 10Å from the residue's C$\alpha$, and concatenate features of the same degree, resulting in a tensor with 940 coefficients. We train several H-AE and H-VAE models, all with $z = 2$; see Sec. A.6.4 for details.

We consistently find that the latent space clusters by amino acid conformations (Fig. A.1), with sharper cluster separations as more training data is added (Fig. A.8

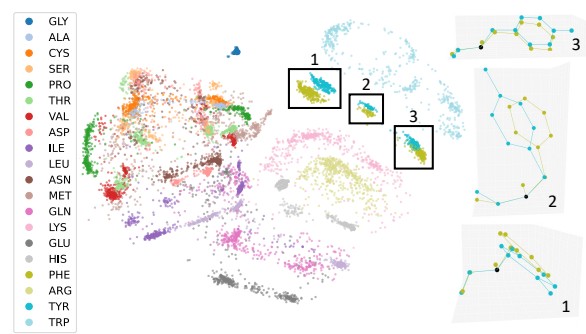

Figure A.1: **H-VAE on amino acids.** H-VAE was trained on 1,000 residues with $\beta = 0.025$ and $z = 2$. The invariant latent space clusters by amino acid conformations. The highlighted clusters for PHE and TYR contain residue pairs with similar conformations; TYR and PHE differ by one oxygen at the end of their benzene rings. We compare conformations by plotting each residue in the standard backbone frame (right); $x$ and $y$ axes are set by the orthonormalized C$\alpha$-N and C$\alpha$-C vectors, and $z$ axis is their cross product.

and A.9). We find that test reconstruction loss decreases with more training data but the reconstruction is accurate even with little training data (from 0.153 Cosine loss with 400 training residues to 0.034 with 20,000); Table A.4. A similar trend is observed for KNN-based classification accuracy of residues (from 0.842 with 400 training residues to 0.972 with 20,000); (Table A.4). Notably, an untrained model, while achieving random reconstruction loss, still produces an informative invariant latent space (0.629 residue type accuracy), suggesting that the forced SO(3)-invariance grants a "warm start" to the encoder. We do not find significant improvements in latent space classification by training with a variational objective, and present ablation results in Table A.5.

## A.4 USING A NON-EQUIVARIANT DECODER: EXTENDED DISCUSSION

Winter et al. (2022) propose to construct group-equivariant autoencoders by using an equivariant encoder that learns an invariant embedding and a group element, and an unconstrained decoder which uses the invariants alone to reconstruct each datapoint in the "canonical" form, before applying the learned group action in the output space. By contrast, for SO(3) we propose to use an equivariant decoder, whereby the learned group element is fed as input to the decoder. Such "unconstrained decoder" procedure can in principle be merged with our equivariant encoder and Fourier-space approach in two ways. For each, we argue in favor of using our equivariant decoder.

**1) Reconstructing the Fourier coefficients of the data.** To apply the learned group element on the decoder's output, the Wigner-D matrices for the data's irreps need to be computed from the group element. Then, the Wigner-D matrices can be used to "rotate" the tensor. This has to be done on-the-fly, and it can be done quickly using functions provided in the e3nn package (Geiger & Smidt, 2022) and by smartly vectorizing operations. We implemented this procedure by using a simple Multi-Layer Perceptron with SiLU non-linearities as a decoder. By using e3nn to compute Wigner-D matrices in batches, and by clever construction of tensor multiplications such that runtime scales linearly with $\ell_{max}$ and is constant with regards to number of channels and batch size, we

achieve models that run with comparable speed to those using our equivariant decoder, and have comparable performance on MNIST (Table A.1). Given the empirical similarities we observe, though on a limited use case, we favor the simplicity and elegance of our equivariant decoder. "Simplicity" because we construct the decoder to be symmetric to the encoder, thus endowing it automatically with similar representational power and without the need to tune an architecture made with different base components. Furthermore, we highlight that our method generates intermediate equivariant representations in the decoder, rather than intermediate invariant representations. These intermediate equivariant representations may be of interest to study in and of themselves.

**2) Reconstructing the data in real space.** In this case, we do not have to compute Wigner-D matrices on-the-fly, since the learned frame can be used directly in the output space as a rotation matrix. However, since the encoder only sees a truncated Fourier representation of the data, which is by construction lossy, while the loss is computed over fine-grained real-space, this model might be too difficult to train. We suspect this would make the model akin to a denoising autoencoder (Vincent et al., 2008) and it might be interesting to analyze, but that would be beyond the scope of this paper. To avoid the denoising effect, we could learn to reconstruct data in real space after an Inverse Fourier Transform (IFT). However, computing the IFT on-the-fly is very expensive and requires a discretization of the input space, to the point of being prohibitive for point clouds. This is not a bottleneck for the Forward Fourier Transform if the cloud is parameterized by Dirac-Delta distributions, i.e., for point clouds, as the integral can be computed exactly (Eq. A.9).

Table A.1: **Performance comparison between our H-(V)AE and H-(V)AE with Winter et al. (2022)'s non-equivariant decoder formulation, on the MNIST-on-the-sphere dataset.** The non-equivariant decoders are constructed as simple MLPs with SiLU non-linearities, with the following hidden layer sizes: [32,64,128,160,256]. We keep the number of parameters approximately the same to make model comparison fair, but we do not tune the architecture of the invariant decoders. All other training details are kept the same (Sec. A.6).

| Method | z | Speed | MSE | Cosine | Purity | V-meas. | LC Class. Acc. | KNN Class. Acc. |
|---|---|---|---|---|---|---|---|---|
| H-AE NR/R | 16 | **1.0** | $9.3 \times 10^{-4}$ | **0.025** | **0.62** | **0.51** | **0.820** | **0.862** |
| H-AE unconst. decoder NR/R | 16 | 1.3 | $1.3 \times 10^{-3}$ | 0.037 | 0.61 | 0.48 | 0.802 | 0.856 |
| H-VAE NR/R | 16 | **1.0** | $1.4 \times 10^{-3}$ | **0.057** | **0.67** | **0.54** | **0.812** | 0.848 |
| H-VAE unconst. decoder NR/R | 16 | 1.3 | $2.1 \times 10^{-3}$ | **0.057** | 0.62 | 0.51 | 0.781 | **0.853** |

## A.5 Implementation details

Without loss of generality, we use real spherical harmonics for implementation of H-(V)AE. We leverage e3nn Geiger & Smidt (2022), using their computation of the real spherical harmonics and their Clebsch-Gordan coefficients.

In our code, we offer the option to use the Full Tensor Product instead of the ETP. Specifically, at each block we allow the users to specify whether to compute the Tensor Product channel-wise or fully-connected across channels, and whether to compute using efficient or fully connected degree mixing.

## A.6 Experimental details

### A.6.1 Architecture specification

We describe model architectures as follows. We specify the number of blocks $B$, which is the same for the encoder and the decoder. We specify two lists, (i) DegreesList which contains the maximum degree $\ell_{max,b}$ of the output of each block $b$, and (ii) ChannelsList, containing the channel sizes $C_b$, of each block $b$. These lists are in the order as they appear in the encoder, and are reversed for the decoder. When it applies, we specify the number of output channels of the initial linear projection $C_{init}$. As noted in the main text, we use a fixed formula to determine $\ell_{max,b}$, but we specify it for clarity.

### A.6.2 MNIST ON THE SPHERE

**Model architectures.** For models with invariant latent space size $z = 16$, we use 6 blocks, DegreesList $= [10, 10, 8, 4, 2, 1]$ and ChannelsList $= [16, 16, 16, 16, 16, 16]$, with a total of 227k parameters.
For models with invariant latent space size $z = 120$, we use 6 blocks, DegreesList $= [10, 10, 8, 4, 2, 1]$ and ChannelsList $= [16, 16, 16, 32, 64, 120]$, with a total of 453k parameters.

**Training details.** We keep the learning schedule as similar as possible for all models. We use $\alpha = 50$. We train all models for 80 epochs using the Adam optimizer (Kingma & Ba, 2017) with default parameters, a batch size of 100, and an initial learning rate of 0.001, which we decrease exponentially by one order of magnitude over 25 epochs. For VAE models, we use $\beta = 0.2$, $E_{\text{rec}} = 25$ and $E_{\text{warmup}} = 35$. We utilize the model with the lowest loss on validation data, only after the end of the warmup epochs for VAE models. Training took $\sim 4.5$ hours on a single NVIDIA A40 GPU for each model.

### A.6.3 SHREC17

**Model architectures.** Both AE and VAE models have $z = 40$, 7 blocks, DegreesList $= [14, 14, 14, 8, 4, 2, 1]$, ChannelsList $= [12, 12, 12, 20, 24, 32, 40]$, $C_{\text{init}} = 12$, with a total of 518k parameters.

**Training details.** We keep the learning schedule as similar as possible for all models. We use $\alpha = 1000$. We train all models for 120 epochs using the Adam optimizer with default parameters, a batch size of 100, and an initial learning rate of 0.0025, which we decrease exponentially by two orders of magnitude over the entire 120 epochs. For VAE models, we use $\beta = 0.2$, $E_{\text{rec}} = 25$ and $E_{\text{warmup}} = 10$. We utilize the model with the lowest loss on validation data, only after the end of the warmup epochs for VAE models. Training took $\sim 11$ hours on a single NVIDIA A40 GPU for each model.

### A.6.4 TOY AMINO ACIDS

**Pre-processing of protein structures.** We sample residues from the set of training structures pre-processed as described in Sec. A.6.5.

**Fourier projection.** We set the maximum radial frequency to $N = 20$ as it corresponds to a radial resolution matching the minimum inter-atomic distances after rescaling the atomic neighborhoods of radius $10.0\text{Å}$ to fit within a sphere of radius 1.0, necessary for Zernike transform.
The channel composition of the data tensors can be described in a notation - analogous to that used by e3nn but without parity specifications - which specifies the number of channels $C$ for each feature of degree $\ell$ in single units $C$x$\ell$: 44x0 + 40x1 + 40x2 + 36x3 + 36x4.

**Model architectures.** All models have $z = 2$, 6 blocks, DegreesList $= [4, 4, 4, 4, 2, 1]$, ChannelsList $= [60, 40, 24, 16, 16, 8]$, $C_{\text{init}} = 48$, with a total of 495k parameters. We note that the initial projection is necessary since the number of channels differs across feature degrees in the data tensors.

**Training details.** We keep the learning schedule as similar as possible for all models. We use $\alpha = 400$. We train all models for 80 epochs using the Adam optimizer with default parameters and an initial learning rate of 0.005, which we decrease exponentially by by one order of magnitude over 25 epochs. For VAE models, we use $E_{\text{rec}} = 25$ and $E_{\text{warmup}} = 10$. We utilize the model with the lowest loss on validation data, only after the end of the warmup epochs for VAE models.
We vary the batch size according to the size of the training and the validation datasets. We use the following (dataset_size-batch_size) pairs: (400-4), (1,000-10), (2,000-20), (5,000-50), (20,000-20). Training took $\sim 45$ minutes on a single NVIDIA A40 GPU for each model.

**Evaluation.** We perform our data ablations by considering training and validation datasets of the following sizes: 400, 1,000, 2,000, 5,000 and 20,000. We keep relative proportions of residue types even in all datasets. We perform the data ablation with H-AE as well as H-VAE models with

$\beta = 0.025$ and $0.1$.

We further perform a $\beta$ ablation using the full (20,000) dataset, over the following choices of $\beta$: $[0(\text{AE}), 0.025, 0.05, 0.1, 0.25, 0.5]$.

For robust results, we train 3 versions of each model and compute averages of quantitative metrics of reconstruction loss and classification accuracy.

For a fair comparison across models with varying amounts of training and validation data, we perform a 5-fold cross-validation-like procedure over the 10k test residues, where the classifier is trained over 4 folds of the test data and evaluated on the fifth one. If validation data is needed for model selection (e.g. for LC), we use 10% of the training data.

### A.6.5   PROTEIN NEIGHBORHOODS

**Pre-processing of protein structures**. We model protein neighborhoods extracted from tertiary protein structures from the Protein Data Bank (PDB) (Berman et al., 2000). We use ProteinNet's splittings for training and validation sets to avoid redundancy, e.g. due to similarities in homologous protein domains (AlQuraishi, 2019). Since PDB ids were only provided for the training and validation sets, we used ProteinNet's training set as both our training and validation set and ProteinNet's validation set as our testing set. Specifically, we make a $[80\%, 20\%]$ split in the ProteinNet's training data to define our training and validation sets. This splitting resulted in 10,957 training structures, 2,730 validation structures, and 212 testing structures. We use pyRosetta (Chaudhury et al., 2010) to assign Solvent Accessible Surface Area (SASA) to every atom.

**Projection details.** We set the maximum radial frequency to $N = 26$ as it corresponds to a radial resolution matching the minimum inter-atomic distances after rescaling the atomic neighborhoods of radius $12.5\mathring{A}$ to fit within a sphere of radius 1.0, necessary for Zernike transform.

**Model architectures.** All models have $z = 64$, 6 blocks, DegreesList $= [4, 4, 4, 4, 2, 1]$, ChannelsList $= [128, 128, 96, 96, 64, 64]$, $C_{\text{init}} = 64$, with a total of 3.5M parameters. We note that the initial projection is necessary since the number of channels differs across feature degrees in the data tensors.

**Training details.** We keep the learning schedule as similar as possible for all models. We use $\alpha = 400$. We train all models for 120 epochs using the Adam optimizer with default parameters, a batch size of 256, and an initial learning rate of 0.002, which we decrease exponentially by one order of magnitude over 40 epochs. For VAE models, we use $E_{\text{rec}} = 40$ and $E_{\text{warmup}} = 40$. We utilize the model with the lowest loss on validation data, only after the end of the warmup epochs for VAE models. Training took $\sim 10$ hours on a single NVIDIA A40 GPU for each model.

### A.6.6   LATENT SPACE CLASSIFICATION

**Linear classifier.** We implement the linear classifier as a one-layer fully connected neural network with input size equal to the invariant embedding of size $z$, and output size equal to the number of desired classes. We use cross entropy loss with logits as training objective, which we minimize for 250 epochs using the Adam optimizer with batch size 100, and initial learning rate of 0.01. We reduce the learning rate by one order of magnitude every time the loss on validation data stops improving for 10 epochs (if validation data is not provided, the training data is used). At evaluation time, we select the class with the highest probability value. We use PyTorch for our implementation.

**KNN Classifier** We use the `sklearn` (Pedregosa et al., 2011) implementation with default parameters. At evaluation time, we select the class with the highest probability value.

### A.6.7   CLUSTERING METRICS

**Purity (Aldenderfer & Blashfield, 1984).** We first assign a class to each cluster based on the most prevalent class in it. Purity is computed as the sum of correctly classified items divided by the total number of items. Purity measures classification accuracy, and ranges between 0 (worst) and 1 (best).

**V-measure (Rosenberg & Hirschberg, 2007).** This common clustering metric strikes a balance between favoring homogeneous (high homogeneity score) and complete (high completeness score) clusters. Clusters are defined as homogeneous when all elements in the same cluster belong to the same class (akin to a precision). Clusters are defined as complete when all elements belonging to the same class are put in the same cluster (akin to a recall). The V-measure is computed as the harmonic mean of homogeneity and completeness in a given clustering.

### A.6.8 ON THE COMPLEMENTARY NATURE OF CLASSIFICATION ACCURACY AND CLUSTERING METRICS

The clustering metrics "purity" and "V-measure" and the supervised metric "classification accuracy" characterize different qualities of the latent space, and, while partly correlated, they are complementary to each other.

Both classes of metrics are computed by comparing the ground truth labels to the predicted labels, and they mainly differ by how the predicted labels are assigned; the clustering metrics use an unsupervised clustering algorithm, while the classification metric uses a supervised classification algorithm to do so. As a result, these metrics focus on different features of the latent space. For example, the clustering metrics are largest when the test data naturally forms clusters with all data points of the same label. While this case can result in a high supervised classification accuracy, clustering is not a necessary condition for high classification accuracy. Indeed, the supervised signal could make the predicted labels depend more heavily on a subset of the latent space features, instead of relying on all of them equally, which is what the clustering algorithm naturally does. Therefore, it is reasonable to conclude that having higher clustering metrics and a lower classification accuracy is a sign that class-related information is more evenly distributed across the latent space dimensions. Overall, the complementary aspect of these metrics makes it necessary to use all of them when comparing the performance of different models in each task.

Table A.2: Evaluation of network performances for MNIST-on-the-sphere using a KNN classifier instead of linear classifier in the latent space. Results are significantly better than when using a linear classifier for models with smaller ($z = 16$) latent space, comparable for the other models.

| Type | Method | z | bw | LC Acc. | KNN Acc. |
|------|--------|---|----|---------|----------|
| Unsupervised | H-AE NR/R | 120 | 30 | 0.877 | 0.875 |
| | H-AE R/R | 120 | 30 | 0.881 | 0.886 |
| | H-AE NR/R | 16 | 30 | 0.820 | 0.862 |
| | H-AE R/R | 16 | 30 | 0.833 | 0.876 |
| | H-VAE NR/R | 120 | 30 | 0.883 | 0.879 |
| | H-VAE R/R | 120 | 30 | 0.884 | **0.895** |
| | H-VAE NR/R | 16 | 30 | 0.812 | 0.848 |
| | H-VAE R/R | 16 | 30 | 0.830 | 0.874 |

Table A.3: QEvaluation of network performances for Shrec17 using a KNN classifier instead of linear classifier in the latent space. Results are better than when using a linear classifier.

| Type | Method | z | bw | Class. Acc. | P@N | R@N | F1@N | mAP | NDCG |
|------|--------|---|----|-------------|-----|-----|------|-----|------|
| Unsupervised + LC | H-AE | 40 | 90 | 0.654 | 0.548 | 0.569 | 0.545 | 0.500 | 0.597 |
| | H-VAE | 40 | 90 | 0.631 | 0.512 | 0.537 | 0.512 | 0.463 | 0.568 |
| Unsupervised + KNN | H-AE | 40 | 90 | **0.672** | **0.560** | **0.572** | **0.555** | **0.501** | **0.599** |
| | H-VAE | 40 | 90 | 0.658 | 0.541 | 0.558 | 0.539 | 0.487 | 0.591 |

Table A.4: Quantitative data ablation results on the Toy amino acids dataset. A random-guessing classifier has an expected accuracy of 0.050.

| # train | H-AE | | | | H-VAE ($\beta = 0.025$) | | | | H-VAE ($\beta = 0.1$) | | | |
|---------|------|-------------|---------|----------|------|-------------|---------|----------|------|-------------|---------|----------|
| | MSE | Cosine loss | LC Acc. | KNN Acc. | MSE | Cosine loss | LC Acc. | KNN Acc. | MSE | Cosine loss | LC Acc. | KNN Acc. |
| 0 | $1.3 \times 10^{-2}$ | 1.015 | 0.409 | 0.629 | $1.5 \times 10^{-2}$ | 0.981 | 0.424 | 0.656 | $1.5 \times 10^{-2}$ | 0.981 | 0.424 | 0.656 |
| 400 | $9.4 \times 10^{-4}$ | 0.153 | 0.586 | 0.842 | $9.8 \times 10^{-4}$ | 0.160 | 0.616 | 0.848 | $1.0 \times 10^{-3}$ | 0.163 | 0.558 | 0.780 |
| 1,000 | $5.9 \times 10^{-4}$ | 0.099 | 0.583 | 0.856 | $6.3 \times 10^{-4}$ | 0.101 | 0.569 | 0.854 | $6.9 \times 10^{-4}$ | 0.113 | 0.564 | 0.844 |
| 2,000 | $4.5 \times 10^{-4}$ | 0.073 | 0.560 | 0.900 | $4.9 \times 10^{-4}$ | 0.081 | 0.554 | 0.905 | $5.5 \times 10^{-4}$ | 0.092 | 0.593 | 0.890 |
| 5,000 | $3.3 \times 10^{-4}$ | 0.053 | 0.629 | 0.940 | $3.3 \times 10^{-4}$ | 0.053 | 0.638 | 0.961 | $4.3 \times 10^{-4}$ | 0.072 | 0.588 | 0.921 |
| 20,000 | $2.2 \times 10^{-4}$ | 0.034 | 0.578 | 0.972 | $2.4 \times 10^{-4}$ | 0.037 | 0.667 | 0.971 | $2.9 \times 10^{-4}$ | 0.047 | 0.662 | 0.966 |

Table A.5: Quantitative data ablation results on the Toy amino acids dataset. Models were trained on the full dataset (# train = 20,000).

| $\beta$ | MSE | Cosine | LC Acc. | KNN Acc. |
|---|---|---|---|---|
| 0 (AE) | $2.2 \times 10^{-4}$ | 0.034 | 0.580 | 0.972 |
| 0.025 | $2.4 \times 10^{-4}$ | 0.037 | 0.666 | 0.971 |
| 0.05 | $2.5 \times 10^{-4}$ | 0.039 | 0.669 | 0.968 |
| 0.1 | $2.9 \times 10^{-4}$ | 0.047 | 0.661 | 0.966 |
| 0.25 | $6.8 \times 10^{-4}$ | 0.132 | 0.597 | 0.854 |
| 0.5 | $1.1 \times 10^{-3}$ | 0.203 | 0.467 | 0.722 |

Table A.6: **Test MSE and Cosine loss for H-(V)AE models trained on MNIST, Shrec17 and Protein Neighborhoods.** MSE and Cosine values are strongly correlated within datasets but not across datasets.

| Dataset | Method | z | bw | MSE | Cosine |
|---|---|---|---|---|---|
| MNIST | H-AE NR/R | 120 | 30 | $6.2 \times 10^{-4}$ | 0.017 |
| | H-AE R/R | 120 | 30 | $6.8 \times 10^{-4}$ | 0.018 |
| | H-AE NR/R | 16 | 30 | $9.3 \times 10^{-4}$ | 0.025 |
| | H-AE R/R | 16 | 30 | $8.9 \times 10^{-4}$ | 0.024 |
| | H-VAE NR/R | 120 | 30 | $1.4 \times 10^{-3}$ | 0.037 |
| | H-VAE R/R | 120 | 30 | $1.4 \times 10^{-3}$ | 0.037 |
| | H-VAE NR/R | 16 | 30 | $2.2 \times 10^{-3}$ | 0.057 |
| | H-VAE R/R | 16 | 30 | $2.1 \times 10^{-3}$ | 0.055 |
| Shrec17 | H-AE | 40 | 90 | $1.8 \times 10^{-4}$ | 0.130 |
| | H-VAE | 40 | 90 | $2.2 \times 10^{-4}$ | 0.151 |
| Protein Neighborhoods | H-AE | 40 | 90 | $6.1 \times 10^{-4}$ | 0.161 |

Table A.7: **Training speed and reconstruction ablations of H-(V)AE models with different Tensor Product rules.** To make comparison fair, models were trained using the same training hyperparameters as described in A.6, and all models were constructed to have comparable number of parameters. Speed was computed as training time and divided by the time of the model using ETP within each dataset. Models with the ETP consistently generates better reconstructions and are usually the fastest. The speed and performance gains of the ETP are most apparent on the Protein Neighborhoods task.

| Dataset | Method | TP-type | $C_{init}$ | ChannelsList | DegreesList | # Params | Speed | MSE | Cosine |
|---|---|---|---|---|---|---|---|---|---|
| MNIST | H-AE | ETP | None | [16,16,16,16,16,16] | [10,10,8,4,2,1] | 227k | **1.0** | $\mathbf{9.3 \times 10^{-4}}$ | **0.025** |
| | H-AE | Full-TP | None | [7,6,6,6,16] | [10,8,4,2,1] | 229k | 1.1 | $1.6 \times 10^{-3}$ | 0.044 |
| | H-AE | Full-TP | None | [5,5,5,5,7,16] | [10,10,8,4,2,1] | 227k | 1.7 | $1.5 \times 10^{-3}$ | 0.041 |
| Shrec17 | H-AE | ETP | 12 | [12,12,12,20,24,32,40] | [14,14,14,8,4,2,1] | 518k | **1.0** | $\mathbf{1.8 \times 10^{-4}}$ | **0.130** |
| | H-AE | Full-TP | None | [5,5,5,8,40] | [14,8,4,2,1] | 518k | **0.9** | $1.9 \times 10^{-4}$ | 0.137 |
| | H-AE | Full-TP | None | [4,3,3,6,6,40] | [14,14,14,8,4,2,1] | 513k | 1.9 | $2.0 \times 10^{-4}$ | 0.142 |
| Protein NBs | H-AE | ETP | 64 | [128,128,96,96,64,64] | [4,4,4,4,2,1] | 3.5M | **1.0** | $\mathbf{6.1 \times 10^{-4}}$ | **0.161** |
| | H-AE | Full-TP | None | [17,24,64] | [4,2,1] | 3.5M | 1.3 | $7.3 \times 10^{-4}$ | 0.199 |
| | H-AE | Full-TP | 56 | [17,17,64] | [4,2,1] | 3.6M | 2.8 | $7.5 \times 10^{-4}$ | 0.207 |

Table A.8: **Mean equivariance error for some of our trained H-(V)AE models.** Errors were computed over 2,000 randomly sampled spherical tensors, each with a randomly sampled rotation. Standard deviation is shown alongside the mean. We also show the mean and standard deviation of the absolute value of the output coefficients, to enable contextualization of the measured equivariance error. The equivariance error due to numerical error (absolute difference in coefficients by rotating input vs. output tensor) is consistently three orders of magnitude lower than the typical absolute value of the coefficients, indicating that equivariance is preserved. The same trend occurs for *untrained* models (not shown here for simplicity).

| Dataset | Method | z | # train | Equiv. Error | Abs. Value |
|---|---|---|---|---|---|
| MNIST | H-AE NR/R | 16 | - | $(2.3 \pm 2.7) \times 10^{-4}$ | $(1.0 \pm 0.9) \times 10^{-1}$ |
| | H-VAE NR/R | 16 | - | $(2.5 \pm 1.9) \times 10^{-4}$ | $(1.3 \pm 1.8) \times 10^{-1}$ |
| Shrec17 | H-AE | 40 | - | $(1.4 \pm 2.4) \times 10^{-5}$ | $(0.4 \pm 1.1) \times 10^{-2}$ |
| | H-VAE | 40 | - | $(1.4 \pm 3.5) \times 10^{-5}$ | $(0.4 \pm 2.2) \times 10^{-2}$ |
| Toy Aminoacids | H-AE | 2 | 1,000 | $(4.7 \pm 3.7) \times 10^{-5}$ | $(2.6 \pm 4.1) \times 10^{-2}$ |
| | H-VAE $\beta = 0.025$ | 2 | 1,000 | $(5.4 \pm 3.9) \times 10^{-5}$ | $(2.8 \pm 4.0) \times 10^{-2}$ |
| | H-AE | 2 | 20,000 | $(9.3 \pm 6.4) \times 10^{-5}$ | $(3.5 \pm 4.7) \times 10^{-2}$ |
| | H-VAE $\beta = 0.025$ | 2 | 20,000 | $(9.0 \pm 4.7) \times 10^{-5}$ | $(3.0 \pm 4.3) \times 10^{-2}$ |
| Protein Neighborhoods | H-AE | 64 | - | $(1.8 \pm 2.0) \times 10^{-5}$ | $(1.0 \pm 1.7) \times 10^{-2}$ |

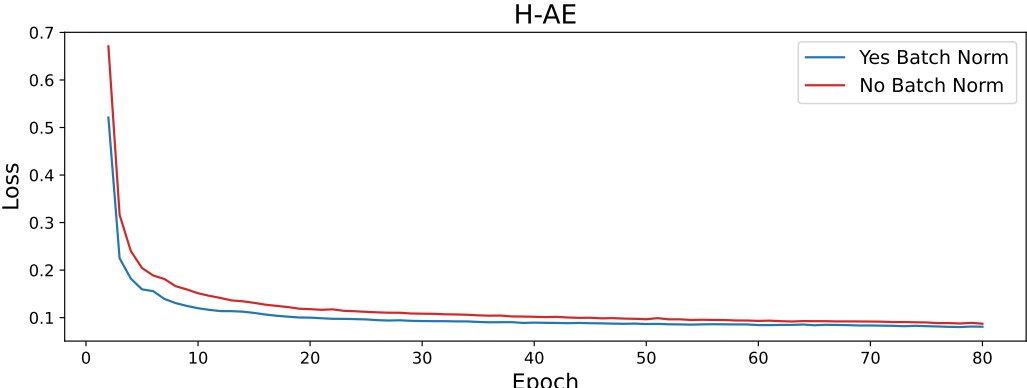

Figure A.2: **Training loss trace of H-AE, with and without Batch Norm, on MNIST-on-the-sphere.** Models were trained with the (NR/R; z = 16; AE) specification. The loss on validation data follows the same trend, but it is not shown for simplicity.

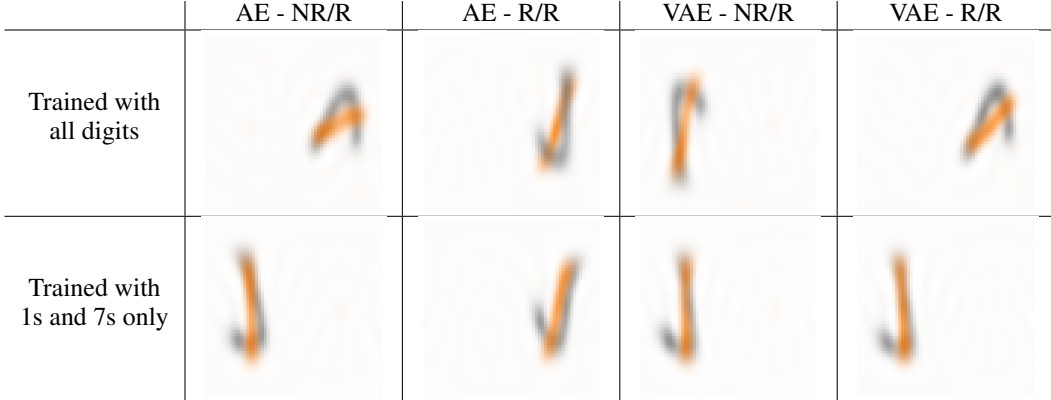

Figure A.3: **Empirical tests to show H-(V)AE learns a canonical frame in the MNIST-on-the-sphere.** For each of the 4 models with $z = 16$, we train a version using only images containing 1s and 7s. For each of the resulting 8 models, we visualize the sum of training images of digits 1 and 7, when rotated to the canonical frame. To achieve the canonical frame we rotate each image such that the frame learned by the encoder is the same for all images, and specifically, we make it correspond to the $3 \times 3$ identity matrix. We compute the sums of images with the same digit, and overlay them with different colors for ease of visualization. We test the hypothesis as whether H-(V)AE learns frames that align the training images such that they maximally overlap; we do so in two ways.

First, if the hypothesis were true, all canonical images of the same digit should maximally or near-maximally overlap - since they have very similar shape - and thus, their overlays would look like a "smooth" version of that digit. Indeed, we find this statement to be true for all models irrespective of their training strategy.

Second, we consider the alignment of images of different digits. We take 1s and 7s as examples given their similarity in shape. If the hypothesis were true, models trained with only 1s and 7s should align canonical 1s along the long side of canonical 7s; indeed we find this to be consistently the case. The same alignment between 1s and 7s, however, does not necessarily hold for models trained with all digits. This is because maximizing overlap across a set of diverse shapes does not necessarily maximize the overlap within any independent pair of such shapes. Indeed, we find that canonical 1s and canonical 7s do not overlap optimally with each other for models trained with all digits.

We note that these tests do not provide proof, but rather empirical evidence of the characteristics of frames learned by H-(V)AE on the MNIST-on-the-sphere task.

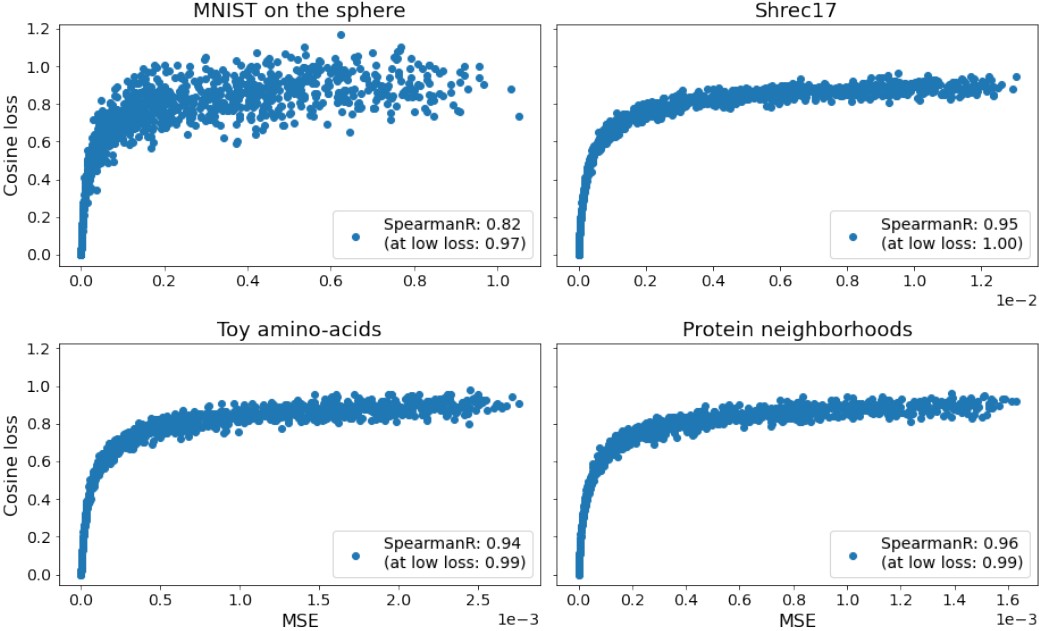

Figure A.4: **Correlation between Cosine loss and MSE values between pairs of random tensors**. For each dataset, we sample a batch of $N = 1000$ tensors with dataset-specific feature degrees and channel sizes, where each coefficient is sampled from a normal distribution. We mimic the normalization step performed in the real experiment and normalize each tensor by the average total norm of the batch. We then generate a "noisy" version of each tensor by adding (normalized) Gaussian noise to each coefficient with standard deviation sampled from a uniform distribution between 0 and some maximum noise level (10 in these plots). This procedure results in $N$ pairs of tensors with varying degrees of similarity between them. We compute the MSE and Cosine loss for all $N$ pairs of tensors and visualize them. The two loss values are well correlated in rank as measured by Spearman Correlation. The correlation is significantly stronger in the regime of reconstruction loss below a Cosine loss of 0.5 (SpearmanR $\sim 0.99$), a value well above the maximum Cosine loss achieved by H-(V)AE in all our experiments. All the p-values for the Spearman Correlations shown in the plot are significant ($< 0.05$).

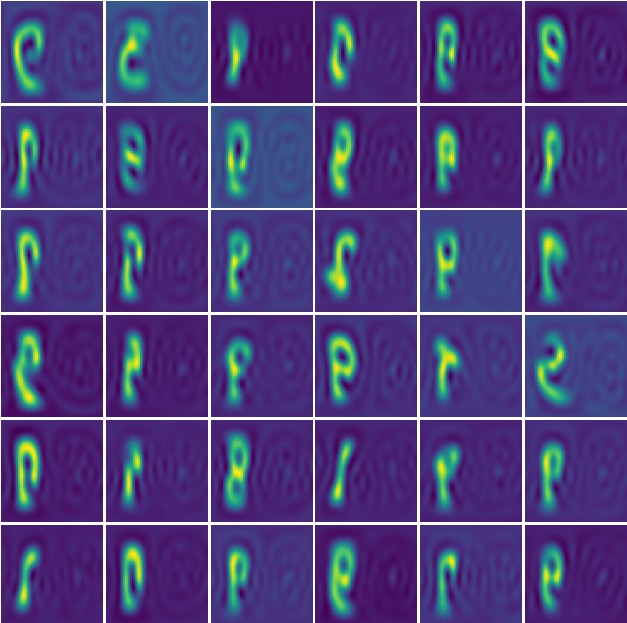

Figure A.5: **Random samples generated by the (NR/R; z = 16; VAE) MNIST-on-the-sphere model.** We sample invariant latent embeddings from the prior distribution (isotropic normal) and feed them to the decoder alongside the canonical frame to generate tensors. We then compute the inverse SFT to map the generated tensor to images in real space. The samples show a wide range of diversity in digit and style.

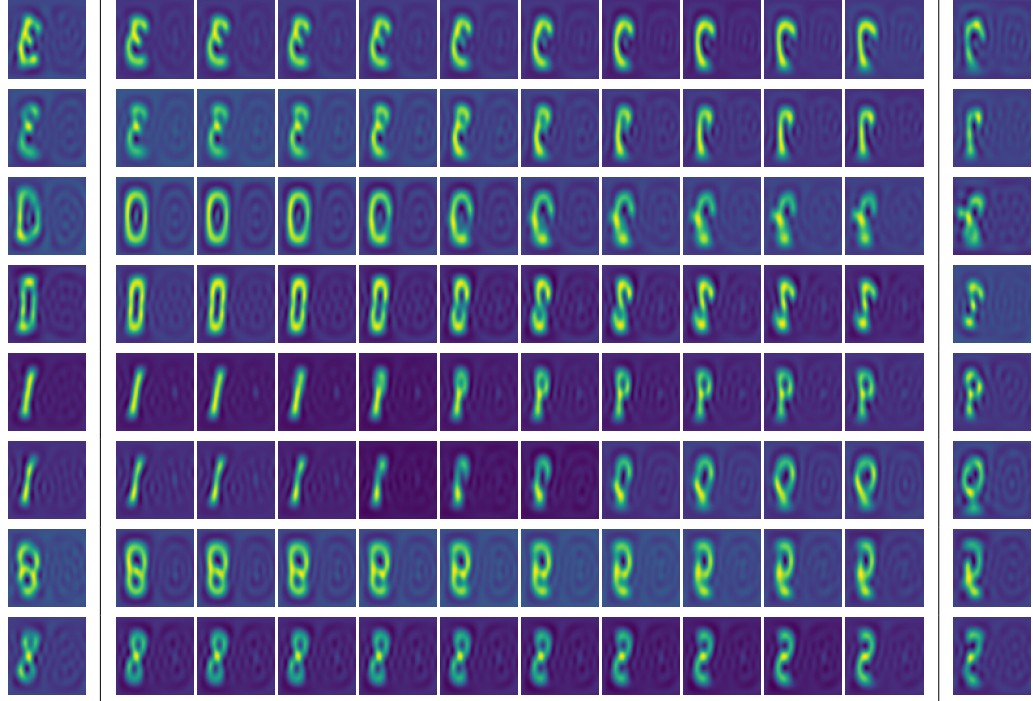

Figure A.6: **Trajectories across the latent space for the (NR/R; z = 16; VAE) MNIST-on-the-sphere model.** We compute pairs of invariant latent embeddings using the model's encoder, and linearly interpolate between them through the latent space. We then feed the interpolated embeddings into the decoder, together with the canonical frame, and compute the inverse SFT to get the image in real space. The left and right columns show the original images (after forward and inverse SFT) rotated to be placed in the learned canonical frame, whereas the center rows show the interpolated images. We can see that all trajectories are smooth, respecting spatial consistency, sign of a well structured latent space.

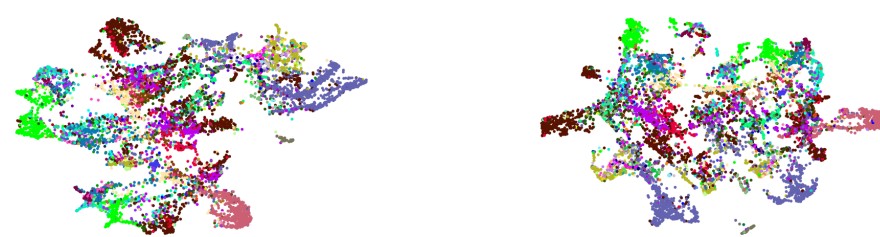

Figure A.7: **2D visualization via UMAP of the invariant latent embeddings of Shrec17 test data learned by H-(V)AE.** Left: H-AE, Right: H-VAE. Points are colored by class (55 classes).

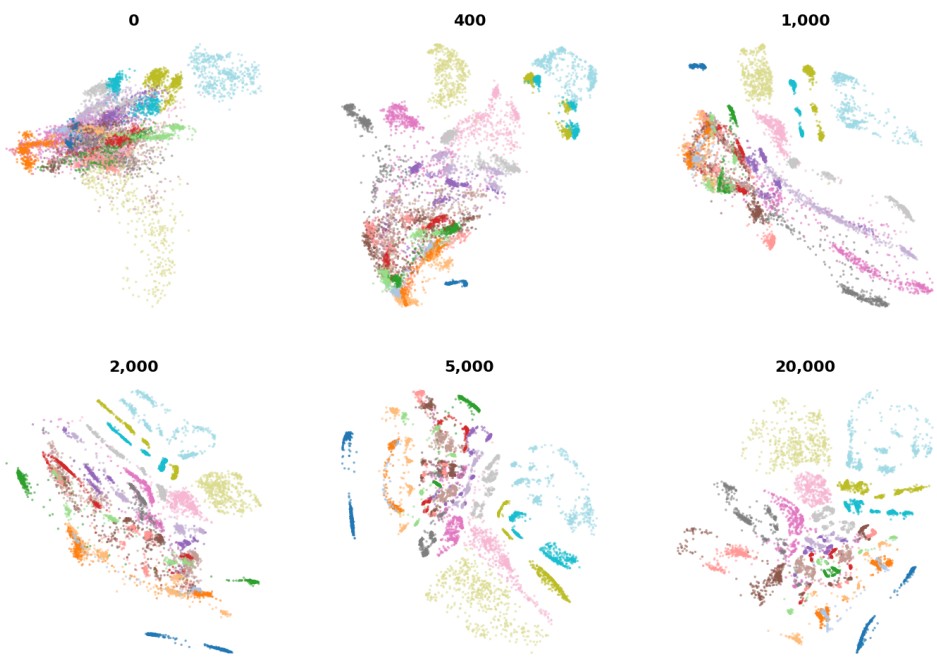

Figure A.8: **Amino acid latent space learned by H-AE.** Visualization of the test data's invariant latent space learned by H-AE trained with varying amounts of the training data. As more training data is added, the separation of clusters containing residues with most similar conformations becomes more distinct. Notably, even with no training data, conformation clusters can be identified.

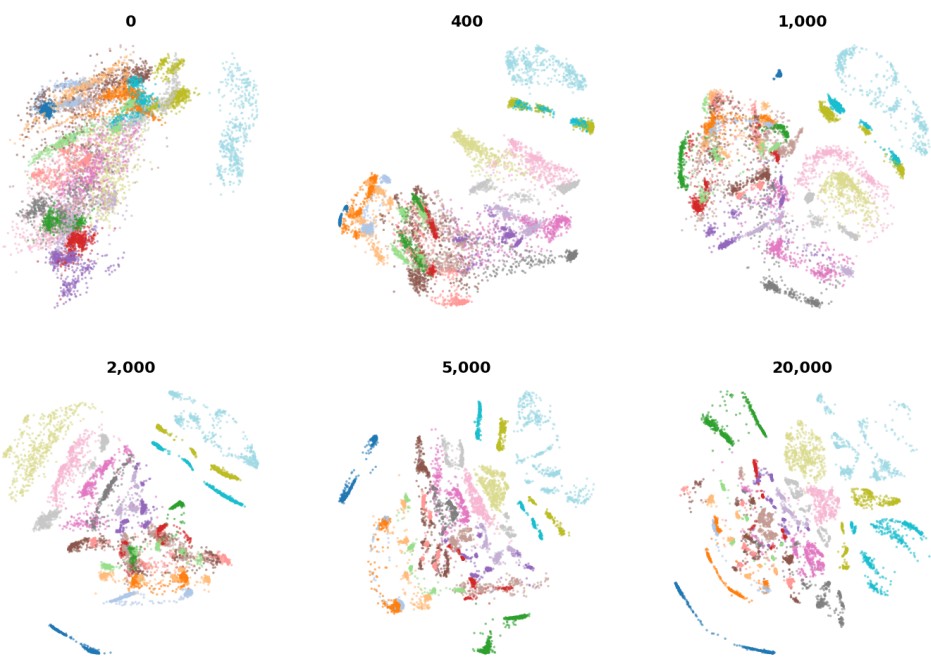

Figure A.9: **Amino acid latent space learned by H-VAE.** Visualization of the test data's invariant latent space learned by H-VAE ($\beta = 0.025$) trained with varying amounts of training data.

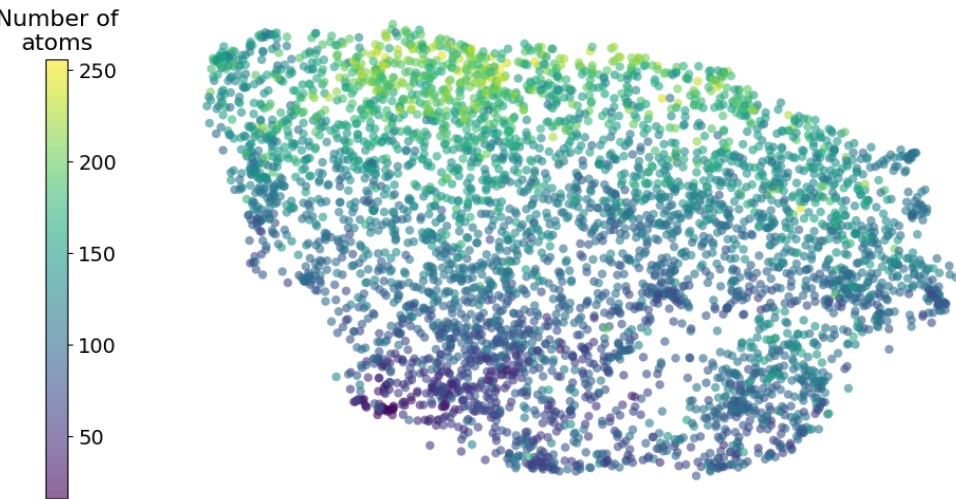

Figure A.10: **Protein neighborhood latent space learned by H-AE annotated by the constituent atoms.** 2D UMAP visualization of the 64-dimensional invariant latent space learned by H-AE, colored by number of the atoms in the neighborhood is shown. A clear gradient can be seen from the bottom of the plot to the top.

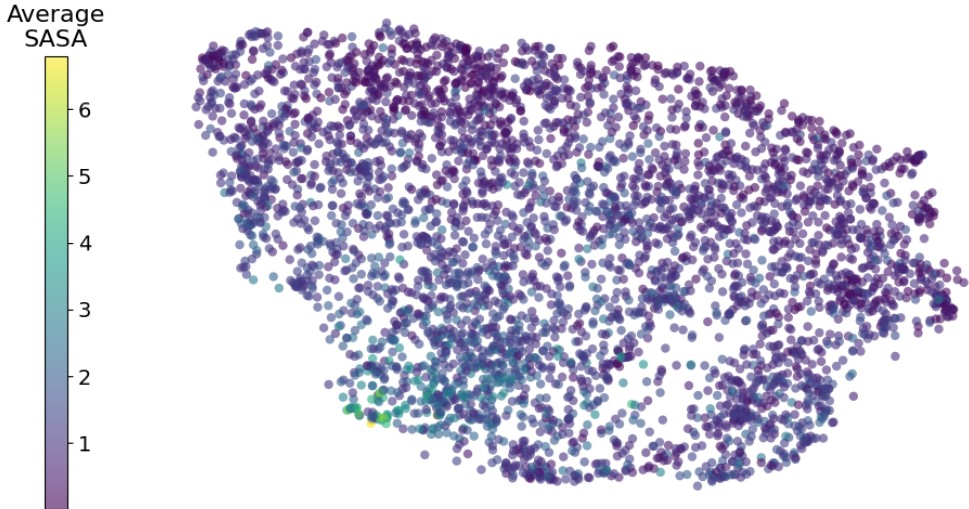

Figure A.11: **Protein neighborhood latent space learned by H-AE annotated by SASA** 2D UMAP visualization of the 64-dimensional invariant latent space learned by H-AE, colored by the average Solvent Accessible Surface Area (SASA) computed for all atoms in each neighborhood is shown. A larger average SASA indicates that a larger proportion of the neighborhood is at the protein's surface. The surface neighborhoods are concentrated in the lower-left side of the map, and the buried neighborhoods are concentrated at the top. Predictably, there is an inverse correlation between the number of atoms (Fig. A.10) and closeness to the surface, as we do not represent a protein's surrounding environment.

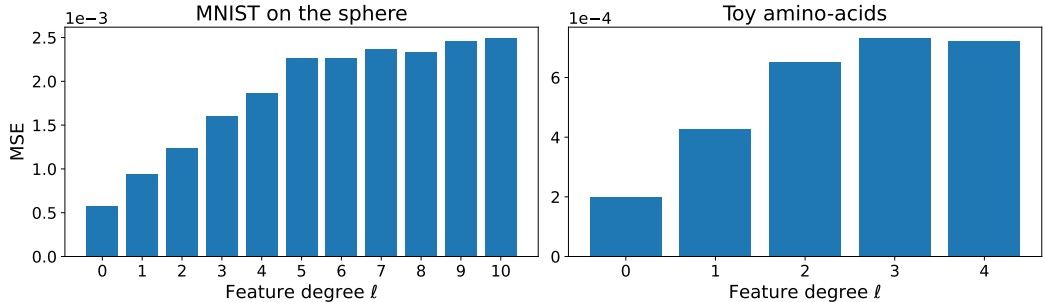

Figure A.12: **Reconstruction loss as a function of feature degree $\ell$.** Test reconstruction loss (MSE) of H-VAE split by feature degree $\ell$, for the MNIST-on-the-sphere (left) and Toy amino acids dataset (right). In both cases, features of larger degrees are harder to reconstruct accurately. The increase in loss is more steep for smaller degrees.

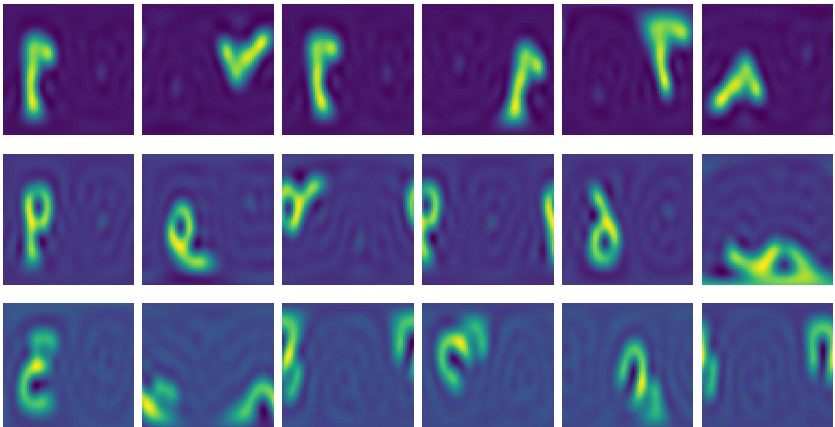

Figure A.13: **Visual proof of the disentanglement in the latent space of MNIST-on-the-sphere.** For each row, the invariant embedding **z** is held fixed, and a different frame (i.e., the rotation matrix) is used. Frames are sampled randomly and differ across rows, with the exception of the first column, which is always the identity frame. Then, **z** and the frame are fed to the decoder and the Inverse Fourier Transform is used to generate the reconstructed spherical image, which is projected onto a plane for the ease of visualization. Modulo the distortions given by projecting the image onto a plane, it is clear that the invariant embedding contains all semantic information, and the frame solely determines the orientation of the image.

