# OpenReview forum: "Holographic-(V)AE: an end-to-end SO(3)-Equivariant (Variational) Autoencoder in Fourier Space"
_ICLR.cc/2023/Conference — Submitted to ICLR 2023_

### Official Review · Reviewer_Wgbf · 2022-10-20

**Confidence:** 2
**Correctness:** 3
**Technical Novelty And Significance:** 2
**Empirical Novelty And Significance:** 1
**Recommendation:** 3

**Clarity, Quality, Novelty And Reproducibility:**

As far as I can see, code is not provided.

I greatly appreciate the effort the authors have made to explain how the steerable features and equivariant transformations are constructed.

Some language could be more precise. For example, in 2.1 I think ‘data that is distributed in 3D space’ means ‘functions on R3’ (with point clouds being sums of delta functions). In 3.3, ‘point clouds of amino acids’ means ‘single amino acids represented as point clouds of atoms’. Use of the word ‘hologram’ seems gratuitous: trying to relate the model to physical holograms (i.e., interference patterns used as diffraction gratings) only confused me.

In 5.2 what are the 6 channels? I expected 3 channels for R, G, B (I'm not familiar with this dataset).

Equation (3): is c channel? Is every h positive? Should each h be squared? I don’t know the term ‘total norm’, is it standard? Can you just use layer norm (Ba et al. 2016)?

3.4, 3.5 MSE loss and cosine metric: I do not understand the claim that you cannot measure goodness of reconstruction using MSE and that the problem is fixed by looking at cosines instead. 2.1 says the Zernike polynomials are a complete orthonormal basis for functions on R3, so MSE in truncated ZFT seems like an OK measure of reconstruction accuracy (it's just MSE ignoring the higher-order Zernike components).


**Strength And Weaknesses:**

I think the idea of the paper is that using steerable features and equivariant models for unsupervised learning should yield good representations for downstream prediction tasks.  It is a reasonable hypothesis, but it wasn’t clear to me if the results really support it. Of the tasks presented, which ones are already better solved in simpler ways? (I’m not familiar with literature on unsupervised classification of 3D shapes).

E(3) equivariant autoencoders already exist, for example in Satorras et al. (2022) https://arxiv.org/pdf/2102.09844.pdf, but I have not previously seen examples using steerable features.  Introducing spherical harmonics etc. adds complexity, which ideally would be justified by results.  Maybe the cited paper by Brandstetter et al. could be a source of ideas about when the extra complexity is worth it.

**Summary Of The Paper:**

The authors propose an SO(3) equivariant variational autoencoder, acting on steerable features. Using standard datasets, they examine the accuracy of reconstruction, and whether the latent representations of objects are useful for object classification and regression tasks.

**Summary Of The Review:**

The model is new, and I believe it may prove to be useful, but I would like the usefulness to be shown empirically. I would like the authors to go beyond 'let's use spherical harmonics in a VAE' to 'here's what it can do that a simpler E(3) equivariant VAE can't do'.

---

> ### Author Response · Authors · 2022-11-19
> **General response to reviewer's questions and concerns**
>
> Weaknesses:
> We thank the reviewer for their comment. However, we believe that the comparison with E(3) equivariant autoencoders is inaccurate in this case, and that our approach, while using steerable features, is arguably simpler.
> Our method focuses on SO(3) rather than E(3) equivariance, and unlike the methods proposed by Satorras et al. and Brandstetter et al., it does not represent data on a graph. Instead, we create a single steerable tensor representing the data. This makes our method amenable to spherical images and centered 3D data, for which E(3) equivariance and graph-based representations would be an overkill, as translation equivariance is not necessary.
>
> The E(3)-equivariant autoencoder in Satorras et al. (2022) learns one latent embedding per node, and only learns to reconstruct the adjacency matrix. In contrast, we are interested in learning a single embedding which, as the authors themselves point out, would be very computationally expensive to compute in the reconstruction phase using a graph-based autoencoder.
> Graph representations scale with the number of nodes, and the complexity of common GNN layers (which is at best a lower bound on the complexity of equivariant layers in Satorras et al. (2022)) scales linearly with the number of edges (https://arxiv.org/abs/1901.00596), and thus at best linearly with the number of nodes. In contrast, our method creates a representation of fixed size independent of the number of nodes, at the price of only considering SO(3) equivariance about a preferred center. We agree that for small molecules like amino-acids a graph representation would be appropriate, but for protein neighborhoods containing 100/200 atoms, scale may become an issue.
>
> As protein neighborhoods have a preferential center in the central residue’s $\alpha$-Carbon, we believe our method can be leveraged, and larger protein structures can be represented by connecting together compact SO(3)-equivariant/invariant representations learned with our method.
>
> Thus, our method is remarkably distinct from existing E(3)-equivariant autoencoders and has different use cases for which it is appropriate and within which it is computationally cheaper.
>
> Question 1:
> We believe we have provided the code as a zip file upon submission.
>
> Q 3:
> We have followed the reviewers suggestions and made the language more precise throughout the manuscript.
>
> Q 4:
> The Shrec17 dataset consists of 3D, colorless model represented as 3D meshes, used as a benchmark for shape retrieval algorithms. As only the 3-dimensional shell (the shape) of the object is relevant for the purpose of shape retrieval, it is a standard practice to project the data onto a sphere to generate lower-dimensional representations that retain enough semantic information for shape retrieval. As stated in Section 5.2, we follow Cohen et al. (2018) to make this projection, which results in spherical images with 6 channels, each containing different information about the surface of the model. We have now added further clarification about this dataset in Section 5.2 of the manuscript.
>
> Q 5:
> We thank the reviewer and apologize for the typo: each $h$ should indeed be squared! $c$ is indeed a channel, as defined in the last paragraph of section 2.1. Our Signal Norm can be seen as a version of Layer Norm that respects SO(3) equivariance.
> The “norm” of a spherical tensor is a standard term, defined as the sum of its squared coefficients, and, we prepended “total” to remind the reader that we compute such norm for the entire spherical tensor (i.e. all channels and all $\ell$’s), as opposed to for each individual channel in each individual $\ell$, as in Batch Norm. Each component contributing to the total norm are further normalized by the size of its irrep ($2\ell+1$), to normalize the contribution of each irrep. To avoid any confusion, we now explicitly define this the “total norm” in the manuscript.
>
> Q 6:
> We thank the reviewer for raising this point. We agree with the reviewer that MSE is generally a reliable metric to use for comparing reconstruction for models within the confines of a single dataset. However, the scale of MSE depends on the characteristics of the data, e.g. the size of the tensors representing the data, and the irreps. In other words, MSE is not a dimensionless measure and therefore, it would be difficult to compare it across datasets. Cosine loss on the other hand, is a quantity that is comparable across datasets as it is dimensionless and interpretable (Section 3.5). A measure with these characteristics is practically useful to guide the tuning of networks because it provides an estimate of how much better the reconstructions can get if a network’s hyperparameters were to be further optimized.
> To demonstrate these effects, we now added Table A.4 and Table A.6, which show MSE and Cosine for our models on all four datasets.
> We refer the reviewer to the updated Sections 3.5 and A.2.5 for discussion on these points.

---

### Official Review · Reviewer_FEZB · 2022-10-21

**Confidence:** 3
**Correctness:** 3
**Technical Novelty And Significance:** 3
**Empirical Novelty And Significance:** 2
**Recommendation:** 5

**Clarity, Quality, Novelty And Reproducibility:**

## Clarify

The paper is very clearly written.  I would have liked some more experimental details in some cases.

## Quality

This seems like a relative high-quality paper, but the experiments seem incomplete to me.  I would like to see more motivation and comparisons to other methods.

## Novelty

Given the number of SO(3)-equivariant methods to date, I was a bit unsure about the claim this is the first SO(3)-equivariant VAE.  Some searching shows that while there are many equivariant generative models, they are mostly not end-to-end equivariant autoencoders (some use diffusion, flows, or are autoregressive).  The one example I found (https://arxiv.org/abs/2205.07309) is conditional and point cloud based making it different from this method.  The method is indeed novel.

## Reproducibility

I did not run the code, but the model description is quite clear and the data appears to be publicly available.



**Strength And Weaknesses:**

## Strengths
- The model is well-described and reasonable.  In contains block diagonal linear layers acting in Fourier space (as found in other SO(3)-equivariant methods, e.g. Thomas '18), tensor product (ETP) non-linearities (Kondor et al., '18) instead of radial non-linearities.  The authors introduce Signal norm to stabilize learning.  The latent space uses a single SO(3) element constructed using Gram-Schmidt and an invariant vector, a design inspired by Winters '22.  While most of these elements originate elsewhere I think they make for a reasonable design here which, so far as I can tell, is the first end-to-end SO(3)-equivariant VAE.
- The part of the experiments I found most interesting was the idea to insert the identity into the decoder and thus see the output in canonical orientation.  From the results, it does seem the model learns to align inputs leading to a disentanglement of frame and identity which is quite interesting and could have potential applications in 3D alignment or classification problems in which it is difficult to classify objects without disentangling pose.

## Weaknesses
- The motivation seems a bit weak to me.  The experiments appear to use relatively small point clouds or spherical signals at a single radial distance.  The latent space embeddings do not seem to lead to better performance in downstream tasks.  They do seem better clustered, have nice interpolations, and can be easily sampled, but why is this desirable and is it not true of previous methods?  I would wager there are good answers to these questions, but I didn't get a strong impression from the paper.  In particular, it would nice to see applications in which orientation-identity disentanglement is critical.
- The paper could use an ablation study.  How do features such as the Fourier space, the Zernicke basis, ETP, signal norm, and a small degree latent space contribute to success?  In particular, most SO(3) methods use simple radial basis functions localized at different radii.  Why is the Zernicke basis superior? Does orthogonality contribute so useful property?
- The paper could use more baselines.  There are many VAEs which could applied in the experimental settings as well as other generative models.  Could some of the claimed features (such as frame-aligned latent vectors), better clustering, etc. be realized by models with different generative paradigms or without equivariance or without equivariance but trained with data augmentation or using some form of canonicalization?  My hypothesis would be the proposed method is better at disentangling the frame, but this could be demonstrated.
- While the theory is sound, the equivariance of the model should be empirically confirmed to verify correct implementation and quantify any error arising from approximation.
- I have a concern/question about the decoder and the size of the latent space.  Given that the input has features with high frequency $l_{max}$ it seems a bit strange to me go down to just an element of SO(3) and invariant features.  The reconstructions do seem to show this is adequate and since the encoder and decoder and complex non-linear functions, I don't think there is any theoretical problem, but it still strikes me as an unusual design choice.   Other works use higher-dimensional mixtures of higher frequency harmonics to encode latent representations of spherical signals.  I'm not sure this is the wrong choice, but I'd like to see the comparison.
- I'm not sure, but here is one version of my concern: the decoder takes $g \in SO(3)$ and $z \in \mathbb{R}^n$ invariant, so $d(g,z) = g d(I, z)$.  However, this uses up the equivariance constraint, so that $d(I,z)$ is unconstrained.  It could thus be a simple MLP with g acting at the end and have the same equivariance property.  So what is gained by using the given architecture?


## Questions
- I'd like a proof signal norm is equivariant.  It seems like summing the components of the vectors would not be, but maybe I've misunderstood the notation.
- Is it possible to use a quantitative measure of for the quality of the random samples from the prior?  It may make it easier to compare to other work.
- Is it not possible to fill in some missing values in table 1 in some way? Why do the supervised methods lack class. Acc.?

**Summary Of The Paper:**

This work presents H-VAE, an end-to-end SO(3)-equivariant VAE.  The input is a radial signal in $R^3$ called $\rho(r,\theta,\phi)$ which is mapped via the Zernike Fourier Transform to a set of coefficients of Zernicke polynomials which are products of spherical harmonics and radial basis functions.  The SO(3)-equivariant encoder and decoder both operate in Zernike Fourier space, using block diagonal linear maps, tensor product, and Clebsch-Gordon decomposition to achieve equivariance.  The latent space is formed by a single element of SO(3) and an invariant vector.  The authors demonstrate in experiments that this results in the model learning a canonical frame for inputs.  That is, inputs are essentially rotated to be aligned to a canonical frame and the rotation is stored along with an invariant encoding of the canonically rotated input.  The authors demonstrate their method in 4 experimental domains, spherical MNIST, SHREC 17 3D model classification, Amino Acids, and Protein neighborhoods.  The evaluation shows the model achieves good clustering by class and meaningful alignments.  The model also has good reconstructions and can be sampled from to produce reasonable outputs.

**Summary Of The Review:**

This paper contains a promising approach and is well-written but the motivation and experiments seem incomplete to me.  My score is tentative and I willing to adjust it based on discussion.

---

> ### Author Response · Authors · 2022-11-19
> **General response to reviewer's question and concerns**
>
> Weakness 1:
> We appreciate the reviewer’s comments on the strengths of our paper. We agree with the reviewer that the motivation behind the work was limited in the original manuscript. We have now extended this discussion in the conclusion.
>
> W2:
> Signal Norm. As we mention in the main text, we found Signal Norm to be necessary to stabilize training, as activations would consistently explode without it, preventing any training.
> ETP. Cobb et al. (2021) empirically show that the use of the ETP enables state-of-the-art classification accuracy and with greater parameter efficiency over competing methods, including using the Full Tensor Product (TP). We empirically verify that the same holds for our H-(V)AE by constructing a series of H-(V)AE models utilizing the Full TP instead of the ETP (Table A.7).
> Zernike. We have extended the discussion of Zernike and the choice of radial bases in Section A.1.5.
>
> W3:
> Autoencoders: We use AEs because we are partly  interested in learning compact low-dimensional latent representations, and because of their ease of implementation. As we now note in the conclusion, a learned compact representation can be used as input for other more complex tasks.
> Canonicalization:  One could always train an autoencoder that extracts a “rotation-invariant” representation by first manually “canonicalizing” each data point in some well-defined coordinate frame, and training an autoencoder on the canonicalized dataset. The characteristics of the invariant latent space will then be conditioned on the criterion used to canonicalize the data. As it is not always obvious how to define such canonicalization on any given dataset, we propose to effectively learn the canonicalization from data. In Figure A.3 we empirically show that H-(V)AE canonicalizes data such that the canonicalized training data points overlap as much as possible. We do not believe this learned canonicalization is necessarily the best solution for any given downstream task. For example, maximizing the overlap between 6s and 9s clearly makes it hard to distinguish them. However, it is convenient as it bypasses the need for manual canonicalization. We leave the task-specific analysis over the impact of canonicalizations to future work.
> Disentanglement: Winter et al. showed, for multiple symmetry groups, that an autoencoder that does not disentangle the equivariant component from the invariant latent space results in a latent space that is less structured by invariant classes. This is due to the fact that in this case the coordinates of the latent space would depend on the data’s orientation, and are not fully invariant. Similarly, Lohit & Trivedi constructed an analogous, poorly-performing baseline specific to SO(3), which they call Vanilla AE. Given the fact that others have systematically studied these effects we do not think that adding additional experiments to our manuscript to explicitly show the effect of disentanglement on the structure of the invariant latent space is novel and necessary for our manuscript.
> Data augmentation: This is very inefficient for 3D rotations (~500 augmentation would be needed!). We are now highlighting this issue in the manuscript (Section 1).
>
> W 4:
> We now show this in Table A.8.
>
> W 5:
> Our framework can be very easily extended to learn compact equivariant embeddings with higher maximum frequency of the spherical harmonics. For this work, we chose to limit the latent space to an element of SO(3) as the sole equivariant component to force the learned invariant embedding to be “complete”, i.e. to contain all invariant information about each datapoint. This design has the following benefits: (1) the learned latent space can be analyzed on their own and clustered based on any invariant property of the data; (2) the invariants can be used out-of-the-box with any non-equivariant algorithm for invariant downstream tasks; (3) the model can be easily made variational in the invariant embeddings alone.
> We speculate that using higher-order harmonics would be an easier learning problem and thus would be able to more easily achieve lower reconstruction error. However, for the reasons outlined above, we believe the use cases of the latent space are different for the two conditions.
>
> W 6:
> Winter et al. propose exactly the outlined procedure, i.e.,  using an unconstrained decoder taking as input only the invariant vector and then applying the group action at the end. We have included a discussion and experiments on using this procedure in the main text (Section 6) and in the Appendix (Section A.4) and argue in favor of using our equivariant decoder.
>
> Question 1:
> Apologies, there was a typo in the equation!
>
> Q 3:
> We show values reported in the papers where each method came from. We also wish to point out that the Shrec17 competition’s standard metrics are strictly more informative than simple classification accuracy and are targeted towards shape retrieval. We now make a note of this in the caption of the Table.

---

> > ### Comment · Reviewer_FEZB · 2022-12-05
> > **Thank you for the response**
> >
> > I appreciate the authors response and edits.  In particular, its nice to have more explanation on motivation, but I still feel this does not come through in the experiments.  I appreciate the additional proofs, corrections, and verifications on equivariance.  I still think more ablations and baseline comparisons would be helpful.  I believe the arguments for why the model should outperform, but I'd like experimental verification.  Similarly, while it may be the case higher order features are not useful, I think a test is in order.  I do really appreciate that the authors including the comparison with the alternate version the decoder from my answer and from prior work.  However, I would tend to think of that method as the simpler of the two.

---

### Official Review · Reviewer_nuL1 · 2022-10-23

**Confidence:** 3
**Correctness:** 2
**Technical Novelty And Significance:** 2
**Empirical Novelty And Significance:** 2
**Recommendation:** 5

**Clarity, Quality, Novelty And Reproducibility:**

The clarity of the paper could be significantly improved. The idea of disentangled equivariant representation learning is interesting, but not novel.

**Strength And Weaknesses:**

**Strength**
1. Inspired on the prior work by Winter et al. (2022), the idea of learning disentangled representations corresponding to the group-invariant embedding of the data as well as its equivariant frame describing its $SO(3)$ rotation is interesting.
2. The authors have tried to conduct extensive experiments to verify the usage of the proposed model.

**Weakness**
1. The writing and structure of the paper can be significantly improved to make the paper clearer. For example, most of the technical terms introduced in Section 2 need further explanation for clarity, and many definitions and explanation of the appendix A.1 should be moved to the main text.
2. Since the theoretical contribution of the paper is limited, one would expect the authors to better explain the architectural design of the proposed model. However, most of the implementation details are buried in the appendix, which makes Section 3 of the main text especially confusing.
3. The authors claim that they achieved state-of-the-art performance in unsupervised clustering and classification. However, the only method to which the paper compared is an arxiv paper by Lohit & Trivedi, 2020. This makes the claim less compelling.
4. Even though the authors explained why MSE is not a "good" measure of the reconstruction, the models are still trained using MSE. It is thus reasonable and fair to compare the results using MSE.
5. It is interesting to see whether the model indeed learns disentangled representation. An easy experiment to conduct is to change the frame in the latent space with a fixed $z$. I am wondering whether this will generate a rotated copy.
6. In table 1, the previous work by Lohit & Trivedi achieves the best result when measured in "purity and V-means", while it achieves the worst result when measured in "classification accuracy". This makes me wonder whether the valuation metric proposed by the authors are convincing.
7. Since Shrec17 is 3D data to begin with, what is the rationale of projecting it to a sphere?

**Summary Of The Paper:**

This paper proposes an end-to-end $SO(3)$-equivariant (V)AE. The idea is to learn, in the spherical Fourier space, an dis-entangled embedding of the input datum -- one corresponding to the invariant embedding describing the datum in a "canonical" orientation, and the other being the equivariant frame describing the datum's orientation. Experiments have been conducted trying to verifying the authors' claim that the proposed model can achieve good unsupervised clustering and classification results on spherical images and protein structures.

**Summary Of The Review:**

Although the idea is interesting, there is a lot of room for the paper to improve.

---

> ### Author Response · Authors · 2022-11-19
> **General response to reviewer's questions and concerns**
>
> Weaknesses 1 and 2:
> We appreciate the reviewer’s point about extending the mathematical foundations of our work. However, due to the limited space we are not able to include the full material from Appendix A.1 in the main text. In the main text, we have summarized the theoretical ideas behind the work and also included key points about the Zernike transform, which are unique to this in the main text. The mathematics behind group equivariant neural networks concepts have been previously introduced in a number of papers and we decided to refer the reader to the Appendix for details on this topic. Nonetheless, we point out the key novel features with regards to unsupervised learning in the main text.
>
> W 3:
> The work on rotationally equivariant/invariant unsupervised learning models is limited. To our knowledge, Lohit & Trivedi’s paper is the only work that learns a 3D rotation-invariant representation from data given in arbitrary orientations. We have now highlighted this fact more clearly in the manuscript (Section 4).
>
> W 4:
> We thank the reviewer for raising this point. We agree with the reviewer that MSE is generally a reliable metric to use for comparing reconstruction for models within the confines of a single dataset. However, the scale of MSE depends on the characteristics of the data, e.g. the size of the tensors representing the data, and the irreps. In other words, MSE is not a dimensionless measure and therefore, it would be difficult to compare it across datasets. Cosine loss on the other hand, is a quantity that is comparable across datasets as it is dimensionless and interpretable (Section 3.5). A measure with these characteristics is practically useful to guide the tuning of networks because it provides an estimate of how much better the reconstructions can get if a network’s hyperparameters were to be further optimized. For example, using Cosine loss we can estimate that our model trained on Shrec17 (best Cosine = 0.130) is not as well optimized as our model trained on MNIST (best Cosine = 0.017). Using MSE, the trend is reversed (1.8e3 vs 6.7e3) due to MSE being dependent on the absolute scale of the tensor coefficients (Figure A.4).
> To demonstrate these effects, we now added Table A.4 and Table A.6, which show MSE and Cosine for our models on all four datasets. We note that, within the same dataset, MSE and Cosine are correlated, which we also show in Figure A.4. Across datasets, however, the correlation does not hold anymore. The typical ranges of MSE values for each dataset, and their corresponding Cosine loss values, are shown in Figure A.4.
> Despite the outlined qualities, and as we mention in the main text, Cosine loss is not suitable for training as it ignores the relative scales of tensors, and is thus degenerate. On the other hand, MSE is a good measure when gauging the model performance for a given dataset, and therefore, it can reliably be used as a metric to minimize when training a model.
> We now clearly state these facts in the manuscript in Sections 3.5 and A.2.5.
>
> W 5:
> We thank the reviewer for their suggestion. Indeed, the model learns disentangled representations, given the enforced architecture. In fact, this is true even without any training, as the learned $\mathbf{z}$ is by construct rotationally-invariant and the learned frame $g \in SO(3)$ is by construct rotationally equivariant.
> We have added a statement in the main text with regards to this feature of the model to make it more clear, as it is the model’s central feature (Section 3.3), and we are also explicitly showing the resulting disentangled representations for the MNIST task in Figure A.13.
>
> W 6:
> We thank the reviewer for pointing this out to us. We have extended the discussion on the differences between these measures in the manuscript (Section A.6.8). Briefly, we argue that these metrics are complementary, and this makes it necessary to use all of them for comparison of the performance of different models in each task.
>
> W 7:
> We should note that for 3D shape data, like Shrec17, only the 3-dimensional shell of the object is relevant for the purpose of shape retrieval, and therefore, using the full volumetric representation is wasteful. Furthermore, if the objects are centered, translation invariance is already taken care of. Thus, projecting the shapes onto the sphere - usually into multiple spherical channels encoding different features of the model’s surface - reduces the dimensionality of the representations while retaining enough semantic information to distinguish shapes. This is a standard practice in computer vision and has been used to benchmark many SO(3)-equivariant classifiers (see Cobb et al. 2021).
>
> On clarity:
> We appreciate the reviewer’s note on clarity. We have now included a more extensive discussion on the possible applications of 3D rotationally equivariant representation learning for other tasks in the Conclusion. We believe this motivation adds perspective to the work and its possible reach.

---

> > ### Comment · Reviewer_nuL1 · 2022-12-06
> > **Thank you for your response**
> >
> > I really appreciate the authors' detailed response. Some of my concerns, such as point 4, 5 and 7 have been clearly addressed. However, I still believe the paper can be further improved in terms of structure and valuation metrics. I am raising my rating to marginally below accept.

---

### Decision · Program_Chairs · 2023-01-20

**Decision:**

Reject

**Justification For Why Not Higher Score:**

The paper is not that innovative, either on the theory or architecture side. The claim of SOTA results has been called into question.

**Justification For Why Not Lower Score:**

n/a

**Metareview: Summary, Strengths And Weaknesses:**

The reviewers seem to agree that this paper presents an interesting idea, and that the architecture is reasonable. However, some found the writing unclear and several cited limited novelty on the architecture and theory side. The paper claims SOTA in clustering, but reviewers found the baselines and ablation to be lacking. Reviewers noted that the complex architecture does not seem to yield to a commensurate improvement in performance. For this reason I recommend to reject the paper in its current form.

**Summary Of Ac-Reviewer Meeting:**

n/a/